# Comparative Analysis of Morphological, Physiological, Anatomic and Biochemical Responses in Relatively Sensitive *Zinnia elegans* 'Zinnita Scarlet' and Relatively Tolerant *Zinnia marylandica* 'Double Zahara Fire Improved' under Saline Conditions

Sara Yasemin [1,2,*] and Nezihe Koksal [3]

1 Department of Horticulture, Faculty of Agriculture, Siirt University, Kezer, 56100 Siirt, Turkey
2 Flanders Research Institute for Agriculture, Fisheries and Food, 9090 Melle, Belgium
3 Department of Horticulture, Faculty of Agriculture, Cukurova University, Balcalı, 01330 Adana, Turkey
* Correspondence: sara.yasemin@siirt.edu.tr

**Abstract:** Salinity is one of the major abiotic stresses in plants. The aim of the present study was to determine the effects of salinity on relatively sensitive *Zinnia elegans* Jacq. 'Zinnita Scarlet' and relatively tolerant *Zinnia marylandica* D.M. Spooner et al. 'Double Zahara Fire Improved' through a comparative analysis of morphological, physiological, anatomic, and biochemical traits. Plants were irrigated at five levels of salt concentrations (0 [control], 50, 100, 150, 200 mM NaCl) for three weeks at one-day intervals in pots under greenhouse conditions. The effects of salinity stress on plant growth parameters, ion leakage, the loss of turgidity, minimum fluorescence ($F_O'$), plant nutrient elements, leaf anatomic parameters, stoma response to the application of light and abscisic acid perfusion, proline content, chlorophyll a, b and total chlorophyll, and carotenoid content were investigated. Differences in the stages and levels of plant response in the relatively sensitive and relatively tolerant cultivar were determined. Proline accumulation appeared to be higher in Double Zahara Fire Improved (D.Za.F.I.) than Zinnita Scarlet (Zi.S.) in the low concentration of salinity. After the application of abscsic acid perfusion to intact leaf surfaces, the stomata of the relatively tolerant cultivar D.Za.F.I. closed earlier (7 min) than Zi.S. (29 min). Ion leakage (32.3%) and Na accumulation (0.9%) in the aerial parts increased dramatically for Zi.S in the 50 mM NaCl treatment. Moreover, values of plant growth parameters, minimum fluorescence ($F_O'$), photosynthetic pigments, and plant nutrient elements all showed a greater decreasing percentage in Zi.S. compared to D.Za.F.I. Stomatal densities for both the abaxial and adaxial surfaces of the leaf decreased in parallel with the increase in salt stress. Palisade parenchyma cell height and leaf thickness values decreased in Zi.S. as salinity increased. In D.Za.F.I., leaf thickness increased by up to 100 mM NaCl while the height of palisade parenchyma cells decreased under high salt stress conditions (100 mM and above). Recommendations for future research include molecular-level evaluations and the study of how to increase salt tolerance in these potentially valuable ornamental cultivars.

**Keywords:** abscisic acid; ion leakage; photosynthetic pigments; plant nutrients; proline; stoma

## 1. Introduction

Abiotic stress in plants can affect growth, quality, and yield. The salinity of soil and water, already an important problem in arid and semiarid regions, is increasing due to climate change, irrigation, and fertilization. Salinity caused by $\geq 4$ dS m$^{-1}$ EC (electrical conductivity) negatively affects all plant growth stages, from seed germination through phenological stages and productivity [1]. Under saline conditions, plants cannot take water from the soil and can accumulate toxic levels of Na$^+$ and Cl$^-$ ions. This osmotic and ionic stress leads to reductions in cell and tissue elongation, and nutritional imbalance and oxidative stress occur [2,3]. The negative effects of these physiological and biochemical

processes cause plant growth to slow. Ratios of Na/K and Na/Ca are important for the homeostasis of plants [4], but calcium and potassium uptake are reduced by sodium uptake. High concentrations of chloride can cause chlorophyll to degrade, which leads to a reduction in photosynthesis [5]. In combination, these effects significantly reduce plant productivity. Plants respond to salinity in two phases: (1) stomata closure and reduced growth and (2) a cytotoxic stage leading to death [6]. Many anatomical characteristics that help the plant cope with salinity is found in salt-tolerant plants [7]. Anatomical modifications of leaf tissues defined as markers of adaptation to abiotic stress include the thickness of the mesophyll and epidermis and the size and density of the stomata [8]. Plants exposed to salinity generally show traits of succulence, thick cuticles, hairs on stems and leaves, salt glands, and sunken stomata. Some reports indicate that salt stress reduces the number of stomata, while others have determined that stomata and epidermis cell numbers increase with the degree of salinity [7,9–11]. Waqas et al. [12] reported that morphological traits, such as stomata density and aperture on the upper and lower sides of leaves, greatly influence the response of the plant to changes in salinity. Abscisic acid (ABA) is a hormone that helps plants respond to environmental stressors such as salinity and drought by regulating stomatal closure and gene expression. Under environmental stress, ABA plays a significant role in regulating the water status of plants by controlling stomatal movement. ABA does this by manipulating the ion fluxes of the guard cells of the stomata, which, in turn, controls the transpiration water loss of the plant and helps it conserve water [13]. In this case, the determination of ABA perception capability with stomata behavior might be important for the detection of the plant's tolerance.

The determination of the salt stress tolerance of ornamental plants is crucial for their selection and use in salty areas, as well as for the use of alternative (saline) water sources. Bedding plants are an important group of ornamental plants that are widely used in landscaping. *Zinnia* (Asteraceae family) is grown as an annual bedding plant used throughout the world during the summer season. It can also be used as a cut flower and potted plant. *Zinnia* has a wide diversity of vegetative characteristics, flower colors, and flower morphology. These features make Zinnia a popular choice for use in landscaping, gardening, and as a cut flower [14].

Studies examining the tolerance of *Zinnia* cultivars against salinity are limited to a few cultivars. Markovic et al. [15] found that *Z. elegans* (MagellanTM Scarlet) was sensitive under saline conditions. They revealed that this cultivar should not be planted in saline areas due to negative effects on plant growth and flower parameters. In previous work [16], we reported on antioxidant defense mechanisms of *Zinnia marylandica* 'Double Zahara Fire Improved' and *Zinnia elegans* 'Zinnita Scarlet' cultivars under saline conditions. These two cultivars can tolerate salinity up to 150 mM NaCl in terms of antioxidant defense, SOD, and CAT enzyme activities, which increased significantly with 150 mM NaCl in both Zinnia species but decreased with 200 mM NaCl. The highest GR enzyme activity was found in 200 mM salinity at *Z. marylandica* 'Double Zahara Fire Improved'. In another work, 20 Zinnia cultivars were screened for germination under saline conditions [17]. While the majority of Zinnia cultivars showed relatively high sensitivity to salt stress at the germination stage, Dreamland Ivory and Dreamland Coral were more tolerant. Macherla and McAvoy [18] determined that *Zinnia elegans* 'Dreamland' could be irrigated with saline water up to 0.5 g L$^{-1}$ NaCl (an EC of 1 dS m$^{-1}$) in a 5-week production cycle without negative effects on plant growth. Manivannan et al. [19] suggested that salt stress led to important decreases in plant growth, biomass, photosynthetic parameters, and pigments and increased the electrolyte leakage potential (ELP), lipid peroxidation, and hydrogen peroxide level. Escalona et al. [20] found that salinity negatively affected plant growth but not flowering in *Zinnia elegans*. *Z. elegans* cv. 'Magellan' was shown to be relatively tolerant to salinity [21]. Niu et al. [22] reported that *Z. marylandica* 'Zahara Yellow', 'Zahara White', 'Zahara Scarlet', 'Zahara Rose Starlight', 'Zahara Fire', and 'Zahara Coral Rose' and *Z. maritima* 'Solcito' cultivars were sensitive to salinity (investigated saline concentrations were 1.4 dS m$^{-1}$ (nutrient solution, control), 3.0, 4.2, 6.0, and 8.2 dS m$^{-1}$

EC). Villarino and Mattson [23] reported that Z. angustifolia 'Star Gold' was sensitive to salinity. Zivdar et al. [24] found that salinity reduced the germination parameters of *Zinnia*, in contrast to Carter and Grieve [25], who showed that marketable flowers could be produced up to 10 dS m$^{-1}$ salinity when using *Z. elegans* cv. 'Benary's Giant Salmon Rose' and 'Benary's Giant Golden Yellow'. Based on these studies of *Zinnia* and the reports that plants react differently to salinity at different developmental stages, we aimed to determine the differences in the effects of salt stress on two cultivars, *Z. elegans* 'Zi.S.' and *Z. marylandica* 'D.Za.F.I', which are considered to be relatively salt-sensitive and relatively salt-tolerant [26], respectively. The study compared the morphological, biochemical, anatomical, and physiological changes in the two cultivars in response to irrigation with saline water.

## 2. Materials and Methods

### 2.1. Plant Material and Experimental Conditions

This study was carried out in the greenhouse of the Department of Horticulture, Cukurova University in Adana, Turkey, during the summer of 2019 (19 May 2019–6 July 2019). In this study, seeds of *Zinnia elegans* Jacq. 'Zinnita Scarlet' (Z.S.) and *Zinnia marylandica* D.M. Spooner et al. 'Double Zahara Fire Improved' (D.Za.F.I.) cultivars obtained from a local seed distributor (Tasaco Farm, Antalya, Turkey) were used as plant material. The average temperature was 32.9/19.7 °C; the average humidity was 54%.

### 2.2. Experimental Design and Treatments

In May 2019, the seeds of Zinnia cultivars were germinated in trays of peat-filled cells, each with a diameter of 3 cm and a height of 4.5 cm. Municipal water was used for irrigation. Once the seedlings had developed four leaves, they were transferred to 2-L plastic pots containing a mixture of peat and perlite in a 2:1 ratio. After a 5-day adaptation period, irrigation with NaCl solutions of different concentrations (0 mM [control], 50 mM, 100 mM, 150 mM, and 200 mM) was initiated and continued for three weeks on a daily basis. Irrigation solutions were prepared using municipal water. The control group and the salt treatment group were provided with macro and micronutrient solutions. The experiment was conducted in a completely randomized design, with four replicates of five plants each, for a total of 20 plants. The salinity treatments were terminated as soon as the first visible symptoms of the damage, such as necrosis on the leaves or differences in plant height, were observed.

### 2.3. Plant Growth Parameters

At the end of the study, the harvested plants were gently washed first with tap water and then with distilled water to remove any remaining peat and perlite. Shoot length, stem diameter, branch number, branch length, leaf width, and leaf length were determined. After plants were divided into roots and shoots (leaves and stems), the fresh weights of plants were recorded. The plants were then dried in an oven, and their dry weight was determined.

### 2.4. Physiological Parameters

2.4.1. Ion Leakage

In order to determine ion leakage of plants, leaf discs (1 cm diameter) were taken from young, fully expanded, and same-type leaves (second and third leaves from the apex). The leaf discs were washed in distilled water and gently blotted dry. Leaf discs (n = 3) were placed in each test tube, and the tubes were shaken for 4 h. The ion leakage in each sample was determined with the EC meter (EC600 model, Extech Instruments) and accepted as the first measurement (EC1). Leaf discs in the same solution were autoclaved; ion leakage at room temperature was determined, and the final measurement (EC2) was accepted. Ion leakage was calculated using the following Equation (1) [27].

$$\text{Ion leakage (\%)} = (\text{EC1}/\text{EC2}) \times 100 \tag{1}$$

### 2.4.2. Loss of Turgidity

Fully expanded and young leaves (second and third leaves from the apex) were used to determine the loss of turgidity under salt stress. First, the fresh weight (FW) of the leaf discs (1 cm) was recorded, and the turgor weight (TW) was determined after the leaf discs were soaked in distilled water for 4 h. The leaves were dried at 70 °C for 24 h, after which the dry weight (DW) was determined [28].

For the calculation of turgor loss, the following Equation (2) was used:

$$\text{Loss of Turgidity (\%)} = [(\text{TW} - \text{FW})/\text{TW}] \times 100 \tag{2}$$

FW: Fresh Weight, TW: Turgor Weight

### 2.4.3. Minimum Fluorescence

The OJIP curve is a graph that shows how Chl fluorescence changes over time when measured on dark-adapted samples. It has four distinct stages: "O", "J", "I", and "P". "O" represents the minimal fluorescence ($F_O'$) caused by energy loss in antenna pigments before it reaches the reaction centers. "J" represents the fluorescence at 2 ms, "I" represents the fluorescence at 30 ms, and "P" represents the highest fluorescence. The OJIP curve is a useful tool for understanding the process of photosynthesis [29]. In this study, we measured the minimum fluorescence ($F_O'$) using a FluorPen FP100 fluorimeter (FluorPen FP100, Photon System Instruments Ltd., Drasov, Czech Republic). At the end of the experiment, readings were made on three leaves from each plant [30].

### 2.4.4. Ion Concentration Analysis

The dried sample of the roots and shoots was used to determine plant nutrient concentrations. The dry materials were ground and digested using the dry digestion method. Sodium ($Na^+$), calcium ($Ca^{2+}$), magnesium ($Mg^{2+}$), potassium ($K^+$), phosphorus (P), Copper ($Cu^{2+}$), manganese ($Mn^{2+}$), iron ($Fe^{2+}$), and zinc ($Zn^{2+}$) concentrations were determined by inductively coupled plasma-atomic emission spectrometry (ICP-AES) [31]. After determining the ion concentrations, the $Na^+/K^+$ and $Na^+/Ca^{2+}$ ratios were calculated. Furthermore, $Cl^-$ concentration was determined with a scientific chloride analyzer—Sherwood [32].

### *2.5. Anatomical Parameters*

#### 2.5.1. Preparation for Stomatal Examination

Leaf samples were taken early in the morning for stoma counting and measurements. The stomatal density and size were measured from replicas of the abaxial and adaxial epidermis of the leaf. In these samples, a piece of nail polish (transparent) was applied to both sides of the lamina. The molds of the leaves were removed by pressing the leaves with tape. The image of the removed molds was taken with a light microscope (Motic, BA210 Trinocular, Xiamen, Hong Kong, China). The Motic Images Plus 2.0 analyzer program was used to determine the number, length, and width of the stomata in the photographed preparations. Stoma width and length on the upper and lower surface of the leaf were measured and determined as μm in the preparation photographs. In the preparations photographed under the microscope, the stomata on the upper and lower surface of the leaf were determined by counting the number of stomata per $mm^2$.

#### 2.5.2. Investigation of Stoma Behaviors against Light Application

In order to verify the behavior of the stomata against ABA perfusion, the responses of the closed stomata in plant leaves to light were investigated. In order to examine the motility of the stoma, the leaf of the intact plant was placed on a vertical microscope table (Leica DM1000 LED, Leica Microsystems GmbH, Wetzlar, Germany) and images were

recorded automatically every 60 s with the camera (Leica ICC500W) which equipped by microscope. In order to test the ability of stomata to open under light and to observe the differences resulting from the application under light, the white light intensity was obtained from the microscope lamp with approximately 300 $\mu$ mol m$^{-2}$ s$^{-1}$. The upper surface of the plant leaves was fixed on an acrylic glass block using double-sided adhesive tape. Stomatal openings were determined from the images taken using the ImageJ program. Then, the measurements taken were graphed. Stoma clearance measurements were carried out on control plants of both cultivars between 08:00 and 16:00 h [33].

### 2.5.3. Investigation of Stoma Behaviors against ABA Perfusion

In order to observe the instantaneous effect of ABA application on stoma behaviors, intact *Zinnia* plants were fixed on the microscope table as described above, and control (pure water) and ($\pm$) ABA (50 $\mu$M) applications were carried out on the leaf surface using the perfusion system as described by [33] and [34]. Aqueous-immersion lenses were used to examine the stomata and apply the ABA solution. During the studies, the working solution (5 mM KCl, 5 mM potassium citrate (pH 5.0), 0.1 mM CaCl$_2$, and 0.1 mM MgCl$_2$] was perfused to allow 0.3 mL of the solution to flow between the leaf and the lens at a continuous flow of 1.5 mL [33,34]. The closing times of the stomata were observed after perfusing ABA from the leaf surface (intact) on the plant in the D.Za.F.I and Zi.S cultivars. After perfusing a solvent of the ABA solution, as a control, the stomata were not closed. This process was carried out to support the ABA. Measurements were carried out at least 8 times on control plants of both cultivars. Stomatal behaviors were investigated using the Image J program.

### 2.5.4. Preparation for Leaf Cross-Section Examinations

The young and fully expanded leaves (third leaf from the apex) were collected at the end of the study. The leaves were cut from the central part of the middle leaflet, near the widest point of each leaf [35]. Paraffin embedding and microtome sectioning were performed in order to obtain leaf cross sections [36]. The samples were transferred to FPA-70 fixation liquid. All the leaf samples were dehydrated using ethanol and tertiary butyl alcohol series (70, 85, 95, 100%) for 4–5 h in each solution, then embedded in paraffin, sectioned longitudinally (10 $\mu$m) with a rotary microtome (Leica RM2245), stained with 0.125% hematoxyline, and mounted in Entellan [37]. Serial sections of 10 $\mu$m thickness were made from the leaf samples with a microtome (Leica RM2245; Leica Microsystems GmbH, Wetzlar, Germany). Leaf thickness, upper and lower epidermis length, the length of the palisade parenchyma, and the width of sponge parenchyma from five predetermined points of five randomly chosen photographs of each leaf lamina cross-section were determined. The measurements were made using image processing software (Motic Images Plus 2.0 analyzer program).

### *2.6. Biochemical Parameters*
### 2.6.1. Proline Content

Plant samples (250 mg) were crushed and ground in a mortar in 5 mL 3% sulfosalicylic acid solution. The ground samples were centrifuged at 3000 rpm for 4 min. After centrifugation, 1 mL of the supernatant part of the samples was taken, and the reaction mixture was prepared by adding acid ninhydrin (1 mL) and glacial acetic acid (1 mL). This prepared mixture reacted in a water bath at 100 °C for 1 h, and the reaction was terminated in an ice bath. The reaction mixture was extracted with 2 mL of toluene while in an ice bath. The spectrophotometer readings were performed at 520 nm irradiance [38].

### 2.6.2. Photosynthetic Pigment and Total Carotenoid Content

The total chlorophyll, chlorophyll a, chlorophyll b, and carotenoid contents were determined according to the Arnon [39] and Lichtenthaler and Wellburn [40] methods. In brief, 150 mg of fresh leaf samples taken from equal points from the plants were crushed and ground in 15 mL of an 80% acetone solution. After the samples were filtered through

filter paper, they were read in a spectrophotometer (Shimadzu, UV 1800, Japan) at 470, 645, and 663 nm wavelengths. The obtained data were calculated according to Arnon [39] and Lichtenthaler and Wellburn [40] using the following Equations (3)–(6)

$$\text{Total chlorophyll} = [(20.2 \times \text{Abs}_{645}) + (8.02 \times \text{Abs}_{663})] \times (-\text{Acetone (mL)})/\text{Leaf (mg)} \quad (3)$$

$$\text{Chlorophyll-a} = [(12.7 \times \text{Abs}_{663}) - (2.6 \times \text{Abs}_{645})] \times (\text{Acetone (mL)})/\text{Leaf (mg)} \quad (4)$$

$$\text{Chlorophyll-b} = [(22.9 \times \text{A}_{645}) - (4.68 \times \text{A}_{663})] \times (\text{Acetone (mL)})/\text{Leaf (mg)} \quad (5)$$

$$\text{Total Carotenoid} = [(1000 \times \text{A}_{470}) - (3.27 \times \text{Chl-a}) - (104 \times \text{Chl-b})]/229 \quad (6)$$

### 2.7. Statistical Analysis

Stress treatment was carried out in a completely randomized experimental design with two factors (salinity and cultivar). Each treatment had four replicates of five plants. Data were subjected to ANOVA, and the means were separated using the LSD (least significant differences) multiple range test at $p < 0.05$. All the statistical analyses were performed using the JMP8.1 Software package.

## 3. Results

### 3.1. Morphological Parameters

The effects of salt stress on the plant growth parameters and physiological, anatomical, and biochemical properties of two *Zinnia* cultivars grown under different salt concentrations (0, 50, 100, 150, and 200 mM) were investigated. The effects of salinity and cultivar on plant growth parameters were important statistically, but we focused on salt and cultivar interaction results to see the different response stages and levels in the study. So, separately, the effects of cultivar and salt on all parameters are given in Tables S1–S10. The effects of the salinity and cultivar interaction were not important in terms of plant growth parameters except for branch length, stem diameter, and root collar diameter (Table 1). Branch length was higher in 0 mM NaCl—D.Za.F.I (9.92 cm) and 50 mM NaCl—D.Za.F.I (9.72 cm) and 0 mM NaCl—Zi.S (9.48 cm). The percentage of the decrease in branch length from the control to 200 mM NaCl was higher in Zi.S (52.6%) than in D.Za.F.I (36.5%). The percentage of the decrease from the control to 200 mM NaCl of root fresh (71.7%), and dry (72.9%) weights, shoot fresh (65.3%) and dry (57.4%) weights were also higher in Zi.S. The appearance of D.Za.F.I and Zi.S under the application of saline irrigation is shown in Figure 1.

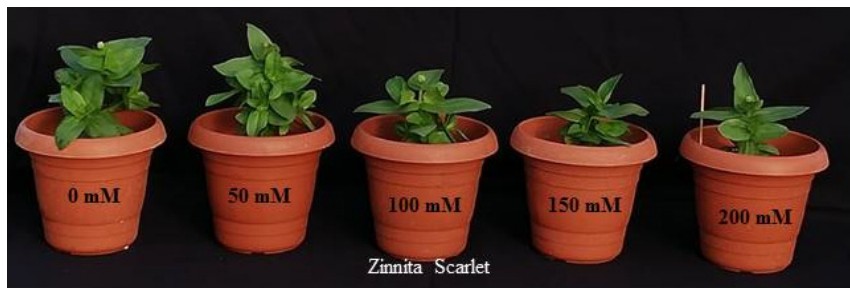

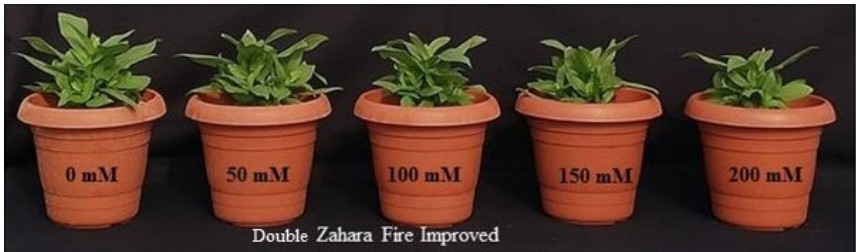

**Figure 1.** The appearence of D.Za.F.I and Zi.S under salt stress.

**Table 1.** Effects of salinity and cultivar interaction on plant growth parameters of D.Za.F.I and Zi.S.

| Plant | NaCl (mM) | Shoot Lenght (cm) | Branch Number (Unit) | Branch Length (cm) | Stem Diameter (mm) | Root Collar Diameter (mm) | Leaf Width (mm) | Leaf Length (mm) | Root Fresh Weight (g) | Root Dry Weight (g) | Shoot Fresh Weight (g) | Shoot Dry Weight (g) |
|---|---|---|---|---|---|---|---|---|---|---|---|---|
| D.Za.F.I | 0 | 15.6 ± 1.4 | 14.9 ± 1.6 | 9.92 + 1.4 a | 5.22 ± 0.2 a | 6.16 ± 0.2 a | 30.4 ± 1.7 | 99.6 ± 6.3 | 6.66 ± 0.7 | 0.48 ± 0.06 | 30.7 ± 0.7 | 3.51 ± 1.0 |
| | 50 | 12.7 ± 1.1 | 14.8 ± 0.9 | 9.72 ± 0.8 a | 4.97 ± 0.5 ab | 5.77 ± 0.3 ab | 28.2 ± 3.2 | 92.5 ± 6.0 | 6.51 ± 0.6 | 0.39 ± 0.02 | 22.1 ± 1.5 | 2.04 ± 0.2 |
| | 100 | 11.4 ± 0.8 | 13.1 ± 1.1 | 8.73 ± 0.4 b | 5.01 ± 0.2 ab | 5.71 ± 0.1 bc | 25.7 ± 3.0 | 84.5 ± 6.9 | 6.11 ± 0.5 | 0.38 ± 0.02 | 19.4 ± 3.1 | 2.14 ± 0.0 |
| | 150 | 10.3 ± 0.5 | 11.9 ± 1.3 | 6.82 ± 0.7 d | 4.44 ± 0.2 c | 5.33 ± 0.1 cd | 24.8 ± 2.5 | 80.0 ± 6.1 | 5.39 ± 0.5 | 0.31 ± 0.04 | 14.9 ± 1.1 | 1.57 ± 0.1 |
| | 200 | 9.9 ± 0.6 | 11.8 ± 0.9 | 6.30 ± 0.7 d | 4.64 ± 0.1 bc | 5.26 ± 0.3 d | 22.3 ± 2.2 | 78.2 ± 3.8 | 4.78 ± 0.8 | 0.28 ± 0.05 | 13.3 ± 2.6 | 1.49 ± 0.2 |
| Zi.S | 0 | 14.9 ± 2.2 | 8.6 ± 2.2 | 9.48 ± 1.2 a | 5.21 ± 0.4 a | 5.63 ± 0.4 bcd | 41.4 ± 5.1 | 74.7 ± 9.8 | 7.90 ± 3.6 | 0.59 ± 0.24 | 24.8 ± 3.2 | 2.70 ± 0.4 |
| | 50 | 13.4 ± 1.2 | 8.4 ± 1.3 | 7.53 ± 1.3 c | 4.82 ± 0.5 abc | 4.71 ± 0.1 e | 39.2 ± 4.1 | 67.1 ± 7.1 | 5.62 ± 2.3 | 0.36 ± 0.11 | 20.0 ± 2.0 | 2.07 ± 0.3 |
| | 100 | 12.8 ± 3.1 | 7.0 ± 1.1 | 6.46 ± 1.0 d | 4.82 ± 0.2 abc | 4.49 ± 0.2 e | 36.0 ± 5.8 | 63.4 ± 7.4 | 3.88 ± 1.6 | 0.30 ± 0.10 | 12.4 ± 4.3 | 1.59 ± 0.3 |
| | 150 | 10.2 ± 1.3 | 7.0 ± 1.2 | 5.06 ± 0.9 e | 4.74 ± 0.1 bc | 4.43 ± 0.1 e | 33.8 ± 3.6 | 60.9 ± 6.7 | 3.99 ± 0.5 | 0.28 ± 0.05 | 12.0 ± 0.7 | 1.55 ± 0.1 |
| | 200 | 10.1 ± 1.0 | 6.7 ± 1.3 | 4.49 ± 0.6 e | 3.64 ± 0.1 d | 3.39 ± 0.1 f | 33.3 ± 5.3 | 58.1 ± 6.6 | 2.23 ± 1.0 | 0.16 ± 0.07 | 8.6 ± 1.4 | 1.15 ± 0.1 |
| LSD | | —NS | —NS | 0.708 ** | 0.445 ** | 0.404 ** | —NS | —NS | —NS | —NS | —NS | —NS |

** $p < 0.01$, —NS: nonsignificant. The differences between the averages were indicated by separate letters.

*3.2. Anatomical Parameters*

3.2.1. Leaf Stomatal Parameters on Abaxial and Adaxial Epiderma

The results of the study revealed that there were statistically significant differences in stomatal parameters between the two cultivars under varying salinity levels, as seen in Table 2. The abaxial stomata width was higher in 0 mM NaCl—D.Za.F.I cultivar (27 μm). The abaxial stomata width decreased slightly in D.Za.F.I as the salt stress increased. A slight increase was observed at 50 mM (20.2 μm) and 100 mM (22.8 μm) of NaCl in the Zi.S. cultivar. The longest abaxial stomata length was found in 0 mM NaCl—D.Za.F.I cultivar (46.3 μm); in this cultivar, abaxial stomata length remained stable under saline conditions. In contrast, the abaxial stomata length of Zi.S decreased as salinity increased. The highest stomatal density was under 0 mM (227 units) and 50 mM (238 units) NaCl for Zi.S, with dramatic decreases observed up to 100 mM NaCl (117 units). The abaxial stomatal density of D.Za.F.I was stable from 0 mM NaCl to 150 mM NaCl, but it dramatically decreased at the 200 mM NaCl (88 units) level. Adaxial stomata width and length were greater at 100 mM (width: 28.5 μm—length: 52.6 μm) and 150 mM (width: 28.0 μm—length: 51.0 μm) NaCl in the D.Za.F.I cultivar. Adaxial stomatal width and length increased at 150 and 200 mM NaCl in both cultivars. Adaxial stomatal density was higher in 0 (85 unit) and 50 (71 unit) mM NaCl in Zi.S cultivar. However, when compared with the control group, the decreasing percentage in adaxial stomatal density was found at 27% in 150 mM NaCl and 25% in 200 mM NaCl treatments in D.Za.F.I plants. The adaxial stomata density decreasing percentage was found at 14%, 13%, and 29% under 100, 150, and 200 mM NaCl treatments, respectively, in the Zi.S. cultivar. Stoma width, length, and density on the abaxial and adaxial epiderma of cultivars are shown in Figure 2.

**Table 2.** Effects of salinity and cultivar interaction on stomatal parameters of D.Za.F.I and Zi.S.

| Cultivar | NaCl (mM) | Abaxial Stomatal Parameters | | | Adaxial Stomatal Parameters | | |
|---|---|---|---|---|---|---|---|
| | | Width (μm) | Length (μm) | Density (unit) | Width (μm) | Length (μm) | Density (Unit) |
| D.Za.F.I | 0 | 27.0 ± 3.0 a | 46.3 ± 4.4 a | 147 ± 32 c | 25.4 ± 2.4 b | 46.7 ± 3.4 b | 85 ± 24 d |
| | 50 | 24.2 ± 2.0 bc | 44.4 ± 3.9 ab | 135 ± 30 c | 22.3 ± 1.8 c | 46.5 ± 2.8 b | 71 ± 11 de |
| | 100 | 24.1 ± 1.7 bc | 43.1 ± 4.6 b | 141 ± 17 c | 28.5 ± 3.9 a | 52.6 ± 3.4 a | 66 ± 10 e |
| | 150 | 24.3 ± 2.0 bc | 43.3 ± 5.6 b | 139 ± 18 c | 28.0 ± 3.2 a | 51.0 ± 3.5 a | 62 ± 15 e |
| | 200 | 24.9 ± 2.4 b | 44.6 ± 5.3 ab | 88 ± 16 d | 25.3 ± 3.7 b | 48.6 ± 3.9 b | 64 ± 14 e |
| Zi.S | 0 | 19.7 ± 2.2 d | 32.1 ± 2.9 d | 227 ± 34 a | 17.9 ± 1.6 d | 30.4 ± 2.1 ef | 175 ± 19 a |
| | 50 | 20.2 ± 2.1 d | 32.4 ± 3.7 d | 238 ± 36 a | 20.5 ± 1.7 c | 32.5 ± 2.2 de | 173 ± 25 a |
| | 100 | 22.8 ± 2.8 c | 36.4 ± 2.4 c | 177 ± 9 b | 18.6 ± 2.1 d | 33.5 ± 2.9 cd | 150 ± 16 b |
| | 150 | 19.8 ± 1.9 d | 30.3 ± 1.9 de | 179 ± 20 b | 21.2 ± 3.0 c | 35.5 ± 5.2 c | 152 ± 28 b |
| | 200 | 19.5 ± 2.7 d | 28.3 ± 3.5 e | 182 ± 25 b | 17.9 ± 2.0 d | 29.4 ± 2.0 f | 124 ± 17 c |
| LSD | | 1.612 *** | 2.802 *** | 23.811 *** | 1.314 *** | 2.290 ** | 17.590 * |

* $p < 0.05$, ** $p < 0.01$, *** $p < 0.001$. The differences between the averages were indicated by separate letters.

In the study, the opening stomatal aperture behavior of the control (0 mM) group plants was investigated by shining light on the closed stomata on the abaxial surface of the leaves. As a result of microscopic examinations, the opening duration of the stomatal aperture of D.Za.F.I and Zi.S. were found to be close to each other under applications of light. The maximum opening of the stomatal aperture was obtained after 36 min in both cultivars (Figure 3a, Supplementary data Video S1). After ABA perfusion, the stomatas of D.Za.F.I (7 min) closed more quickly than Zi.S (29 min) (Figure 3b, Supplementary data Videos S2 and S3). In Figure 4, sample views in two cases with the stomata open and closed in the control group of D. Za.F.I and Zi.S cultivars are presented.

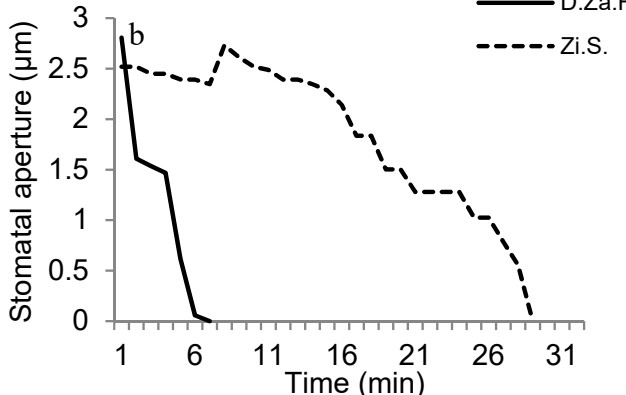

**Figure 2.** The effects of salt stress on stoma width, length, and density on abaxial and adaxial epiderma of D., Za., F., I, and Zi.S. Scale bar: 30 μm.3.2.2. Stoma Behavior against Light Application and ABA Perfusion.

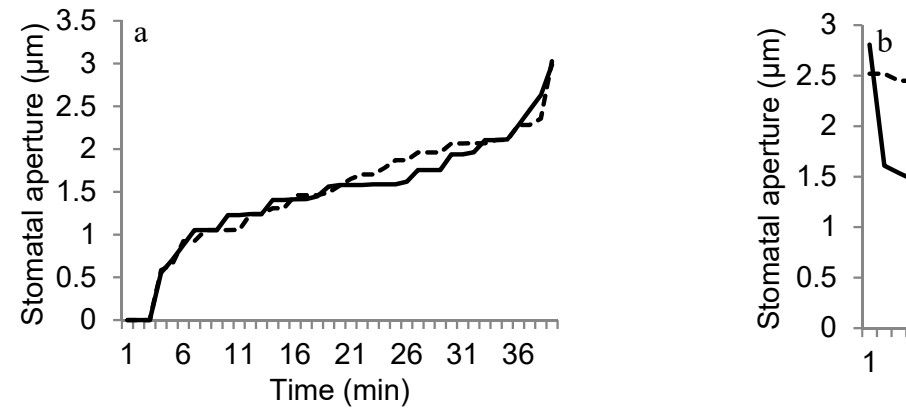

**Figure 3.** The change in stomatal aperture of D.Za.F.I and Zi.S under light application (**a**) and ABAperfusion (**b**).

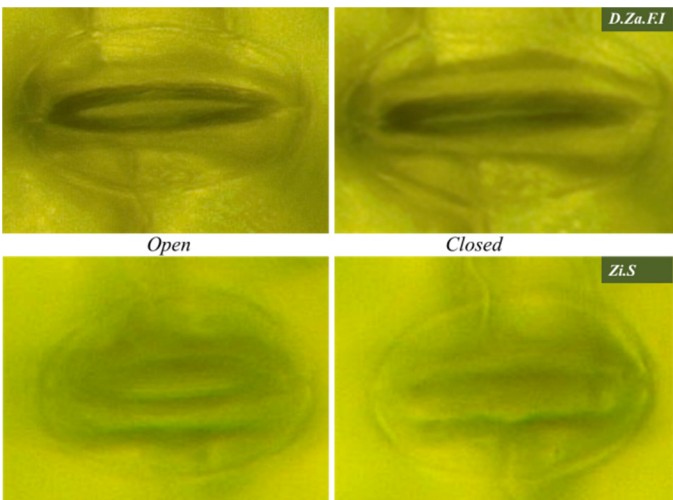

**Figure 4.** The view of open and closed stomata of intact control (0 mM NaCl) plants of D.Za.F.I and Zi.S.

### 3.2.2. Leaf Cross-Sections

The study found that the salinity levels had a significant effect on leaf anatomical parameters, with the exception of the length of the lower surface epidermal cells in both cultivars (Table 3). The highest leaf thickness (LT) value was found at 0 mM NaCl (359.7 μm)—Zi.S. cultivar, and the lowest values were at 150 mM (246.2 μm) and 200 mM (247 μm) NaCl—D. Za.F.I cultivar. When the control (0 mM NaCl) and 200 mM NaCl treatments were compared, leaf thickness decreased by a total of 9% in D.Za.F.I and 28% in Zi.S. The length of the palisade layer (LPL) was lower in 150 mM (54 μm) and 200 mM NaCl (54.4 μm)—D.Za.F.I. The width of spongy parenchyma was higher in the 100 mM NaCl—Zi.S cultivar (23.9 μm). The lowest values were obtained from 100 mM, 150 mM, and 200 mM NaCl—D.Za.F.I cultivar. The length of adaxial epidermal cells was higher in 100 mM NaCl—D.Za.F.I cultivar (19.8 μm). The aspects of leaf cross-sections of D.Za.F.I and Zi.S are presented in Figure 5. As salinity increased, glandular hairs (trichomes) were observed on the leaves of the *Z. marylandica* Double Zahara Fire Improved cultivar (data not presented) (Figure 6).

**Table 3.** Effects of salinity and cultivar interaction on stomatal parameters of D.Za.F.I and Zi.S.

| Cultivar | NaCl (mM) | LT (μm) | LAbEC (μm) | LPL (μm) | WSPC (μm) | LAdEC (μm) |
|---|---|---|---|---|---|---|
| D.Za.F.I | 0 | 272.8 ± 21 c | 25.5 ± 5 | 73.0 ± 5 b | 17.4 ± 4 de | 15.6 ± 3 cd |
| | 50 | 270.5 ± 12 c | 24.8 ± 6 | 73.0 ± 8 b | 15.7 ± 2 ef | 15.1 ± 3 d |
| | 100 | 292.1 ± 23 b | 33.5 ± 8 | 57.8 ± 7 d | 14.2 ± 4 f | 19.8 ± 6 a |
| | 150 | 246.2 ± 12 d | 28.1 ± 5 | 54.0 ± 5 e | 14.4 ± 2 f | 18.1 ± 3 abc |
| | 200 | 247.0 ± 26 d | 27.6 ± 6 | 54.4 ± 8 de | 14.5 ± 3 f | 18.3 ± 5 ab |
| Zi.S | 0 | 359.7 ± 22 a | 20.5 ± 3 | 84.1 ± 7 a | 22.0 ± 3 ab | 11.3 ± 6 e |
| | 50 | 303.1 ± 10 b | 23.9 ± 4 | 72.2 ± 4 b | 20.0 ± 3 bc | 18.2 ± 7 ab |
| | 100 | 303.4 ± 36 b | 33.1 ± 5 | 82.2 ± 4 a | 23.9 ± 5 a | 17.2 ± 5 bcd |
| | 150 | 302.5 ± 57 b | 28.5 ± 5 | 80.5 ± 7 a | 19.2 ± 3 cd | 15.7 ± 8 bcd |
| | 200 | 255.6 ± 33 cd | 25.0 ± 4 | 67.5 ± 4 c | 17.6 ± 4 de | 15.2 ± 7 d |
| LSD | | 17.996 *** | —NS | 3.826 *** | 2.180 *** | 2.611 ** |

** $p < 0.01$, *** $p < 0.001$, —NS: nonsignificant, LT: leaf thickness, LAbEC: length of abaxial epidermal cells, LPL: length of palisade layer, WSPC: width of spongy parenchyma cells, LAdEC: length of adaxial epidermal cells. The differences between the averages were indicated by separate letters.

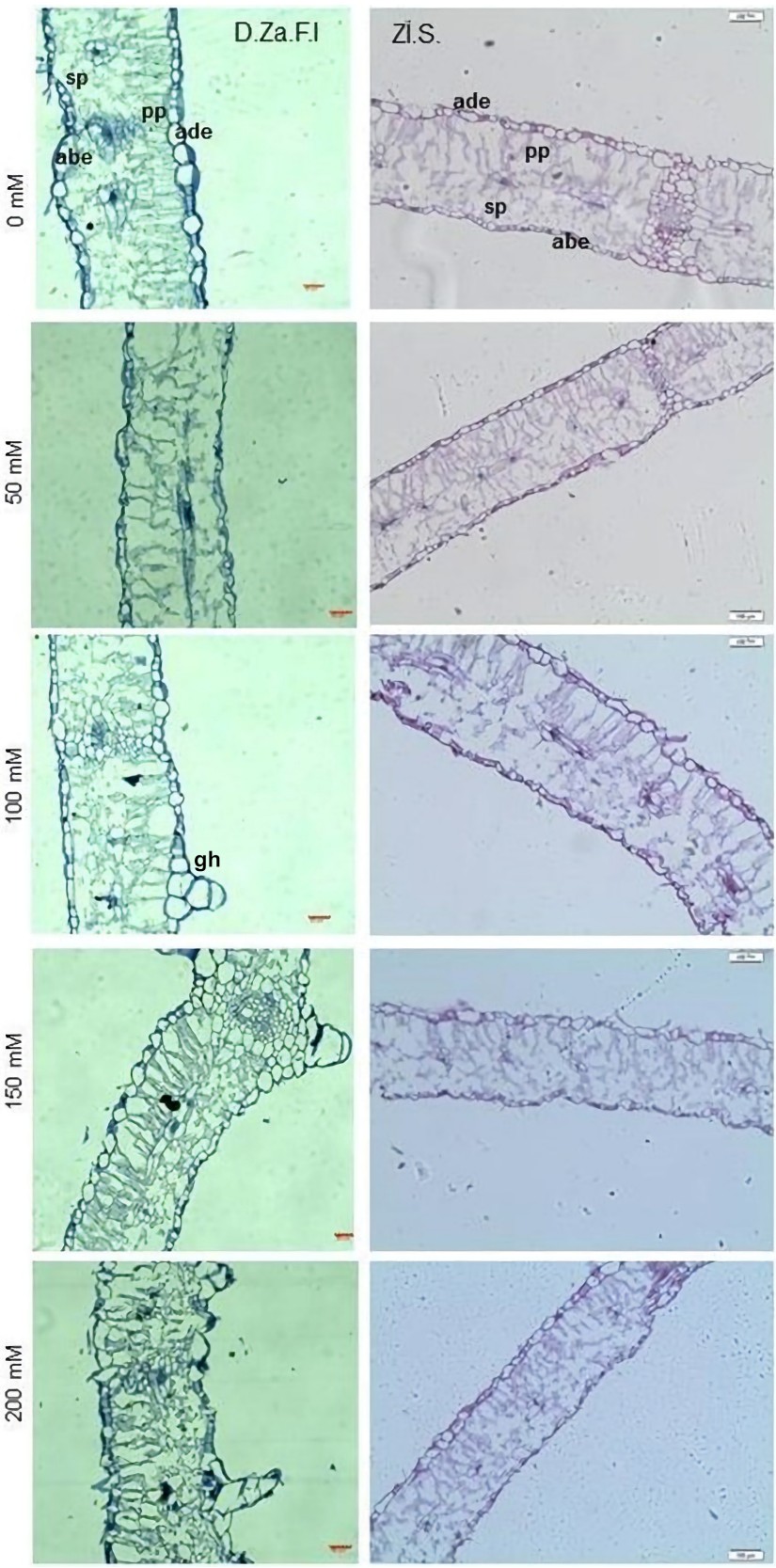

**Figure 5.** Cross sections of Zi.S and D.Za.F.I under salinity stress. ade: adaxial epiderma, abe: abaxial epiderma pp: palisade parenchyma, sp: spongy parenchyma, gh: glandular hair (trichome) (scale bar for Zi.S.: 50 μm, scale bar for D.Za.F.I: 100 μm).

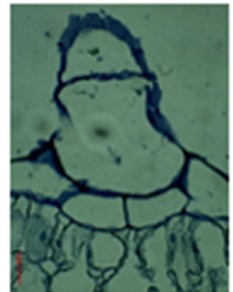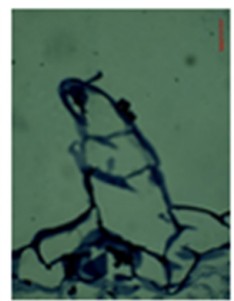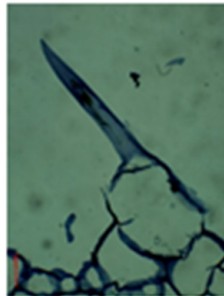

**Figure 6.** Glandular hairs on D.Za.F.I plants under saline conditions, scale bar: 100 µm.

*3.3. Physiological and Biochemical Parameters*

Ion leakage, proline content, $F_O{}'$, and photosynthetic pigments were all affected by salt stress and cultivar interactions, as shown by the statistical analysis (Figure 7). Ion leakage had the lowest value in the control group of D.Za.F.I (25.8%), and the control group of Zi.S (27.9%) and 50 mM NaCl of D.Za.F.I (28.1%) followed. The ion leakage increased drastically in 100 mM NaCl of Zi.S cultivar (50.9%); the increasing rate from the control to 100 mM NaCl was 82%. The ion leakage increase was in 150 mM NaCl for D.Za.F.I cultivar (41.7%) (Figure 7A). The loss of turgidity increased in both cultivars, but interaction effects did not differ between the cultivars (Figure 7B). While proline content increased in the 50 mM NaCl level in D.Za.F.I (7.14 mg/g FW), it was 3.74 mg/g FW in the same concentration of Zi.S. The proline increased in 100 mM NaCl in Zi.S (12.3). A greater proline content was found in the D.Za.F.I cultivar (20.7 mg/g FW) compared to Zi.S (14.7 mg/g FW) under 200 mM NaCl. (Figure 7C). The minimum fluorescence value ($F_O{}'$) had important differences in both cultivars under different salt concentrations. While $F_O{}'$ was stable and slightly increased in 100 mM NaCl in D.Za.F.I, it decreased as the salinity increased in Zi.S (Figure 7D). It was found that the content of photosynthetic pigments in both cultivars was affected by salinity, and this effect was statistically significant. Photosynthetic pigments were stable and slightly increased by 150 mM NaCl in D.Za.F.I. As the salinity increased, photosynthetic pigment contents decreased in Zi.S (Figure 7E–H).

Based on the data presented in Table 4, it was determined that there were no statistically significant differences in the root content of P, K, Ca, Mg, Fe, Cu, Mn, and Zn among the Zinnia cultivars that were evaluated when subjected to salt stress conditions.

**Table 4.** Effects of salinity and cultivar interaction on plant nutrient elements in roots of D.Za.F.I and Zi.S.

| Plant | NaCl (mM) | P | K | Ca | Mg | Fe | Cu | Mn | Zn |
|---|---|---|---|---|---|---|---|---|---|
| | 0 | 0.59 ± 0.10 | 1.86 ± 0.6 | 0.25 ± 0.02 | 1.36 ± 0.2 | 278.2 ± 72 | 40.1 ± 2.5 | 24.4 ± 5 | 117.0 ± 19 |
| | 50 | 0.61 ± 0.12 | 2.16 ± 1.0 | 0.29 ± 0.09 | 1.12 ± 0.2 | 270.2 ± 76 | 41.5 ± 1.5 | 24.4 ± 2 | 166.2 ± 40 |
| D.Za.F.I. | 100 | 0.70 ± 0.05 | 0.97 ± 0.9 | 0.34 ± 0.06 | 1.05 ± 0.4 | 252.7 ± 6 | 41.1 ± 1.2 | 28.8 ± 11 | 150.8 ± 65 |
| | 150 | 0.79 ± 0.01 | 1.42 ± 0.4 | 0.32 ± 0.05 | 0.88 ± 0.2 | 165.8 ± 18 | 38.2 ± 2.9 | 23.0 ± 6 | 132.3 ± 14 |
| | 200 | 0.61 ± 0.11 | 0.75 ± 0.4 | 0.31 ± 0.06 | 0.92 ± 0.3 | 215.4 ± 42 | 42.3 ± 0.5 | 28.5 ± 12 | 163.7 ± 54 |
| | 0 | 0.55 ± 0.15 | 1.51 ± 0.5 | 0.34 ± 0.08 | 1.12 ± 0.3 | 237.8 ± 90 | 44.7 ± 7 | 63.7 ± 23 | 86.9 ± 15 |
| | 50 | 0.55 ± 0.02 | 1.45 ± 0.9 | 0.42 ± 0.09 | 1.30 ± 0.1 | 277.3 ± 63 | 56.9 ± 24 | 68.2 ± 32 | 86.8 ± 13 |
| Zi.S. | 100 | 0.49 ± 0.15 | 0.62 ± 0.0 | 0.53 ± 0.07 | 0.83 ± 0.2 | 260.6 ± 110 | 45.5 ± 8 | 57.2 ± 28 | 94.1 ± 24 |
| | 150 | 0.52 ± 0.25 | 1.28 ± 0.9 | 0.50 ± 0.14 | 0.77 ± 0.3 | 255.2 ± 74 | 48.4 ± 7 | 57.9 ± 24 | 95.4 ± 6 |
| | 200 | 0.39 ± 0.19 | 0.85 ± 0.1 | 0.70 ± 0.30 | 0.45 ± 0.3 | 191.6 ± 52 | 59.9 ± 13 | 43.8 ± 25 | 84.6 ± 11 |
| LSD | | —NS | —NS | —NS | —NS | —NS | —NS | —NS | —NS |

—NS: nonsignificant.

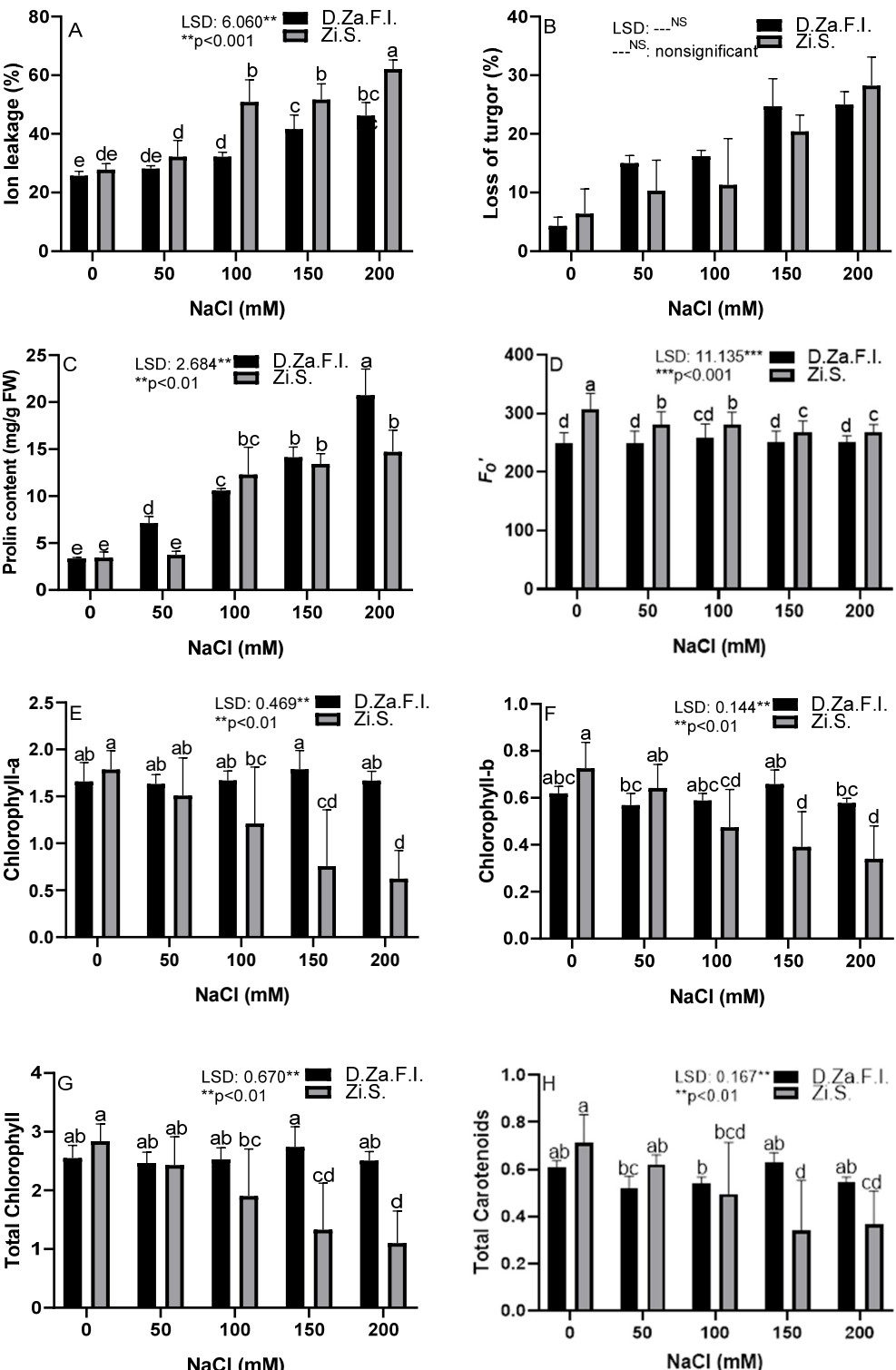

**Figure 7.** The change in ion leakage (**A**), turgidity (**B**), proline content (**C**), minimum fluorescence ($F_O'$) (**D**) Chlorophyll-a (**E**), Chlorophyll-b (**F**), total chlorophyll (**G**), and total carotenoids (**H**) of D.Za.F.I and Zi.S under different levels of salinity. —$^{NS}$: nonsignifcant, ** $p < 0.01$ *** $p < 0.001$. The differences between the averages were indicated by separate letters.

Under saline stress conditions, the concentrations of P, K, Ca, Mg, Fe, Cu, Mn, and Zn in the aerial parts of both cultivars exhibited similar trends of decline or enhancement, with no statistically significant variation between the cultivars. (Table 5). However, the cultivar and salinity interaction effect were important for the N content. The highest N content was in 200 mM NaCl—D.Za.F.I (4.29), 50 mM NaCl—Zi.S (4.48), and 100 mM NaCl—Zi.S (4.46) combinations. The N content in the D.Za.F.I increased with increasing salinity. However, it dramatically decreased to 150 mM (1.87) and 200 mM NaCl (0.30) in the Zi.S.

**Table 5.** Effects of salinity and cultivar interaction on plant nutrient elements in aerial parts of D.Za.F.I and Zi.S.

| Plant | NaCl (mM) | N | P | K | Ca | Mg | Fe | Cu | Mn | Zn |
|---|---|---|---|---|---|---|---|---|---|---|
| D.Za.F.I. | 0 | 2.68 ± 0.37 bc | 0.77 ± 0.03 | 5.30 ± 0.9 | 0.57 ± 0.03 | 1.54 ± 0.03 | 203.9 ± 31 | 46.1 ± 5 | 204.3 ± 15 | 174.2 ± 15 |
| | 50 | 2.88 ± 0.60 bc | 0.79 ± 0.01 | 6.19 ± 0.4 | 1.03 ± 0.30 | 1.82 ± 0.11 | 188.1 ± 27 | 43.1 ± 1 | 206.1 ± 41 | 180.9 ± 18 |
| | 100 | 3.33 ± 0.78 ab | 0.73 ± 0.05 | 6.12 ± 0.1 | 0.86 ± 0.13 | 1.74 ± 0.15 | 162.9 ± 14 | 42.6 ± 6 | 226.9 ± 17 | 168.2 ± 27 |
| | 150 | 3.94 ± 0.24 ab | 0.76 ± 0.03 | 5.84 ± 0.8 | 1.05 ± 0.11 | 1.87 ± 0.15 | 149.9 ± 11 | 39.9 ± 2 | 239.2 ± 11 | 179.6 ± 27 |
| | 200 | 4.29 ± 0.64 a | 0.71 ± 0.03 | 5.84 ± 0.7 | 1.11 ± 0.20 | 1.91 ± 0.36 | 232.6 ± 10 | 42.4 ± 2 | 273.8 ± 10 | 157.1 ± 23 |
| Zi.S. | 0 | 3.84 ± 0.36 ab | 1.01 ± 0.07 | 4.10 ± 1.4 | 0.46 ± 0.07 | 1.91 ± 0.17 | 144.0 ± 19 | 38.7 ± 3 | 223.3 ± 36 | 200.4 ± 65 |
| | 50 | 4.48 ± 0.79 a | 1.14 ± 0.09 | 4.96 ± 0.6 | 0.55 ± 0.12 | 1.91 ± 0.39 | 143.3 ± 6 | 38.7 ± 4 | 209.7 ± 49 | 142.6 ± 60 |
| | 100 | 4.46 ± 0.07 a | 1.21 ± 0.11 | 4.43 ± 0.6 | 0.67 ± 0.06 | 1.99 ± 0.20 | 151.1 ± 18 | 44.2 ± 5 | 298.4 ± 50 | 165.4 ± 69 |
| | 150 | 1.87 ± 1.99 c | 1.05 ± 0.16 | 4.58 ± 0.5 | 0.79 ± 0.08 | 1.94 ± 0.10 | 144.3 ± 6 | 38.9 ± 3 | 272.5 ± 18 | 225.1 ± 12 |
| | 200 | 0.30 ± 0.29 d | 1.09 ± 0.17 | 4.97 ± 1.9 | 0.95 ± 0.19 | 1.87 ± 0.37 | 138.9 ± 8 | 38.7 ± 2 | 241.6 ± 23 | 201.8 ± 54 |
| LSD | | 1.361 *** | —NS | —NS | —NS | —NS | —NS | —NS | —NS | —NS |

\*\*\* $p < 0.001$, —NS: non-significant. The differences between the averages were indicated by separate letters.

The effects of root and aerial part Na, Cl content, Na/K, and Na/Ca ratio under salinity are shown in Figure 8. The root Na content of both cultivars increased as salinity increased, but the interaction was non-significant (Figure 8A). While the content of Na in the aerial parts of D.Za.F.I. increased at 100 mM NaCl, Na in the aerial parts of Zi.S. increased to 50 mM NaCl and increased dramatically at 100 mM (Figure 8B). The Na and Cl content in the aerial parts of the sensitive Zi.S cultivar was higher than that in the tolerant D.Za.F.I cultivar. Increased Na ions were found in both cultivars under salinity conditions, with a higher increase percentage observed in the sensitive cultivar Zi.S. Compared to the control (0 mM NaCl) treatment, the Na+ aerial content (0.23%) in D.Za.F.I increased by 78% at 50 mM NaCl, 171% at 100 mM, 892% at 150 mM, and 1135% at 200 mM NaCl. In Zi.S, the Na+ content was increased by 263% at 50 mM, 880% at 100 mM, 1238% at 150 mM, and 1355% at 200 mM NaCl compared to the control (0.25%). Although the effects of interaction salinity and cultivar on the ratio of Na/K (Figure 8C) and Na/Ca (Figure 3E) in the root were non-significant, ratios increased with salinity. The change in the Na/K and Na/Ca ratios was significant in aerial parts (Figure 8D,F). Both ratios increased with salinity, but the increases in Zi.S were higher than in D.Za.F.I. The content of Cl in the root and shoot increased with salinity in both cultivars, but interaction effects were non-significant (Figure 8G,H). The Cl accumulation in the root and aerial parts increased in both D.Za.F.I and Zi.S cultivars with increasing NaCl concentrations. In D.Za.F.I, the increase was 136% at 50 mM, 200% at 100 mM, 209% at 150 mM, and 264% at 200 mM NaCl compared to the control (1.1%). In Zi.S, the increase was 120% at 50 mM, 160% at 100 mM, 280% at 150 mM, and 300% at 200 mM NaCl, compared to the control (1.0%).

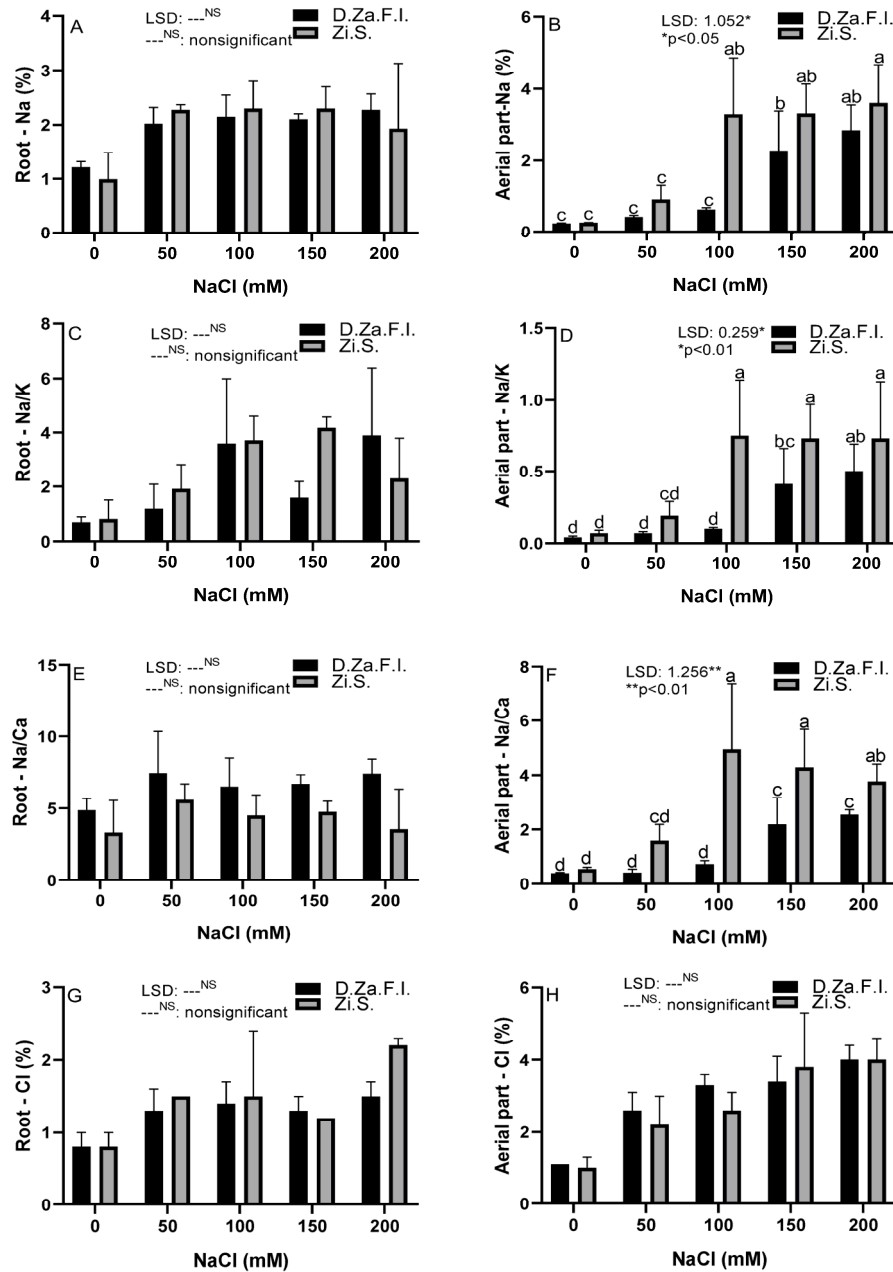

**Figure 8.** The change in root-Na (**A**), aerial part-Na (**B**) root Na/K (**C**) aerial part-Na/K (**D**), root Na/Ca (**E**), aerial part-Na/Ca (**F**), root Cl (**G**), aerial part Cl (**H**) content of D.Za.F.I, and Zi.S under different levels of salinity. The differences between the averages were indicated by separate letters.

## 4. Discussion

Salt (NaCl) taken up by plants does not directly affect plant growth; it first affects the turgor state, photosynthesis, and some enzyme activities [41]. In addition, Munns [41] reported that growth retardation begins with a decrease in soil water potential and then continues with specific effects such as salt damage which primarily affects old leaves due to the excessive accumulation of salt ions in the cell wall or cytoplasm. Salt ions accumulating in old leaves accelerate cell death and prevent the transport of carbohydrates and growth hormones to growth tissues. This excessive accumulation of salt ions slows plant growth as a result of the decrease in the rate of photosynthesis and the formation of growth-inhibiting metabolites. In this study, shoot length, branch number, branch length, stem diameter, root collar diameter, leaf width, leaf length, root fresh weight, and root dry weight values

decreased in general as the salt stress level increased. Many previous studies on ornamental plants show that plant growth parameters decreased with salt stress [21,22,42–48].

In relatively sensitive Zi.S leaf thickness, the length of palisade parenchyma cells, and the width of sponge parenchyma cells decreased as salt stress increased. The length of lower and upper epidermis cells increased up to 100 mM NaCl and then decreased as salt stress increased. The leaf thickness of the D.Za.F.I cultivar, which is relatively tolerant to salt stress, reached a peak value at 100 mM and decreased as NaCl concentrations increased. The length of the palisade parenchyma cells and the width of the sponge parenchyma cells decreased as salt concentrations increased. The lengths of the lower and upper epidermis cells increased with the increasing salt stress. Navarro et al. [49] reported that in *Arbutus unedo*, no change was observed in the palisade parenchyma of the first layer in the leaves exposed to salt stress compared to the control group, while the size of the palisade parenchyma cells in the second layer increased significantly in parallel with the increase in stress. They also found a significant decrease in the intercellular spaces of the sponge parenchyma. Fernandez-Garcia et al. [50] reported that increased leaf thickness could be observed in *Lawsonia inermis* L. plants under highly saline conditions. Acosta-Motos et al. [51,52] investigated leaf anatomy in *Myrtus communis* and *Eugenia myrtifolia* plants under salt-stress conditions. While they observed no anatomical changes in the palisade parenchyma in the *Myrtus communis* plant, they found a decrease in the cells of the sponge parenchyma and an increase in the intercellular spaces. However, the size of palisade parenchyma cells increased significantly. Likewise, Gomez-Bellot et al. [53] reported an increase in leaf thickness along with an increase in the palisade parenchyma cells in *Vibirnum tinus* plants. Hameed et al. [54] revealed that 200 mM NaCl increased succulence in *Imperata cylindrica* (L.) Raeuschel plants. It is known that the increase in leaf thickness and in the palisade parenchyma as salt stress increases helps to facilitate the diffusion of carbon dioxide ($CO_2$) and its progress through the layer, helping the chloroplasts to reach higher rates of $CO_2$ in the palisade parenchyma. Such anatomical changes are particularly important for the maintenance and advancement of the photosynthetic performance of plants under stress situations, which tend to reduce the stomatal opening but also help the plants cope with salt stress [51,52]. The study by Li et al. [55] emphasizes the crucial role of leaf anatomy in regulating the balance between water and $CO_2$ diffusion during drought conditions. The research suggests that by understanding the relationship between leaf anatomy and drought stress, it may be possible to develop more drought-tolerant crops in the future. Plants have the ability to alleviate the negative effects of salt stress by adjusting the density and size of their stomata. This is thought to be an adaptive mechanism that allows plants to respond to changes in environmental factors such as temperature and water availability [12,55,56]. Waqas et al. [12] suggested that the increase in stomatal density in quinoa plants under salinity stress may be a result of the shrinkage of pavement cells, which provides more surface area for $CO_2$ assimilation. This allows the plant to maximize its $CO_2$ uptake, continue its photosynthesis process, and improve its water usage efficiency in response to salinity. Stomatal density is a trait that is dependent on the species, the duration, and the intensity of salinity. An increase in stomatal density leads to an increase in the ion requirements of charge balancing and maintaining osmotic potential at the plasma membrane of guard cells. This is thought to contribute to salinity tolerance. We found that salinity had a negative impact on stomatal density in this study. The stomatal aperture was observed to decrease more rapidly in the cultivar and was relatively tolerant to salinity (D.Za.F.I) compared to the cultivar that was relatively sensitive to salinity (Zi.S).

Leaf water potential and changes in osmotic potential depend on the osmotic potential of the root environment and the amount of stress the plant is experiencing [57]. The plant's stress level can thus be measured by using some parameters that reveal the plant's water content under salt stress conditions. The results of the present study indicate that turgor loss in the leaves of the two *Zinnia* cultivars was low when irrigated with 100 mM NaCl. As salinity increased to 150 and 200 mM, the loss of turgidity increased. Salt stress did not affect

the relative water content of *Pelargonium* [58] and calla lily [59]. In rose, Carvalho et al. [44] reported that the relative water content decreased as salt stress increased. The maintenance of relative water content and turgidity under salt stress conditions is associated with an increase in Ca and Mg accumulation in leaf tissues as well as Na accumulation in plants [58]. Salt stress in plants leads to damage to the cellular membrane. In the present study, increased an ion leakage was observed as salt stress levels increased in both salt-sensitive and salt-tolerant *Zinnia* cultivars. Cell damage rates were found to be higher in the relatively sensitive Zi.S than the relatively tolerant D.Za.F.I. Trivellini et al. [60] reported that ion leakage increased under 200 mM salt stress conditions in *H. rosa-chinensis*.

Salt stress can negatively impact the process of photosynthesis in plants. In the short-term, high salt concentrations can cause the plant's stomata to close, reducing the rate of photosynthesis. This can lead to a halt in plant growth within a few hours of exposure [61]. In the long term, salt accumulation in young leaves can lead to a decrease in chlorophyll and carotenoid levels, which are essential for the process of photosynthesis [22,62–67]. Many studies have shown that photosynthesis, specifically the PSII (Photosystem—II) process, is negatively impacted by salt stress [50,68–70]. The present study revealed that the minimum fluorescence ($F_O'$) decreased in the relatively sensitive Zi.S as salt stress increased while it was preserved in relatively tolerant D.Za.F.I under the same conditions. The chlorophyll-a and total chlorophyll content of cv. D.Za.F.I, which is relatively tolerant to salt stress, did not change as the salt stress increased; a slight decrease in chlorophyll-b content was observed. In contrast, the sensitive cv. Zi.S showed a significant decrease in chlorophyll-a, chlorophyll-b, and total chlorophyll contents under salt stress. Chlorophyll content decreased in salt-sensitive plants [65,71]. Mukarram et al. [72] reported that chlorophyll fluorescence, chlorophyll content, and plant growth were minimized under high salt concentrations (240 mM) in lemongrass (*Cymbopogon flexuosus*). Several reports show that the total chlorophyll content decreases with salt stress: Vernieri et al. [73] in *Acacia cultriformis*, *Callistemon citrinus*, *Carissa edulis microphylla*, *Gaura lindheimeri*, *Jasminum sambac*, *Westringia fruticosa*; Eom et al. [74] in *Alchemilla mollis*, *Nepeta faassenii*, *Phlox subulata*, *Solidago cutleri*, *Thymus praecox*; Lee and van Iersel [75] in *Chrysanthemum morifolium*; Bahadoran and Salehi [76] in *Polyanthes tuberosa*; and Cantabella et al. [77] in *Stevia rebaudiana*. We further found that the total carotenoid content of the plants of both cultivars decreased as salt stress increased. When the total carotenoid content of the plant leaves in the highest salt stress condition (200 mM NaCl) was compared with the content of the control group, and the decrease percentage (9%) in the D.Za.F.I was quite low; it was found to be quite high in the Zi.S (48%). Carotenoids are a crucial class of biochemicals, such as antioxidants, which protect membrane lipids against oxidative stress induced by environmental stressors such as salt stress, thus promoting plant health and survival [78,79]. Furthermore, carotenoids can interconvert and thus contribute to increasing tolerance under stress conditions. [80,81].

Another common plant response to salt stress is an increase in intracellular osmotic regulators. Among organic osmolytes, proline is one of the most important and effective substances. In addition to its role as an osmoprotectant, proline helps plants cope with a variety of environmental stresses, as it has antioxidant properties and acts as a molecular chaperone to protect the structure of biological macromolecules during water loss from the cell [82,83]. In general, proline accumulation in salt-tolerant plants increases after exposure to salt stress. In this study, we found that the amount of proline increased in two *Zinnia* cultivars that were relatively sensitive and tolerant, respectively. However, in the salt-tolerant D.Za.F.I cultivar, proline content increased as soon as the plant was exposed to salt (50 mM NaCl). Similarly, a 20% increase in the leaf proline content was determined in *Eugenia myrtifolia* L., which was tolerant to 8 dS m$^{-1}$ NaCl. [51]. In addition, Bizhani et al. [19] in *Zinnia elegans* 'Magellan' cultivar, Bres et al. [58] in *Pelargonium*, and Li et al. [84] in *Crysahthemum* reported a large increase in proline content. Kumar et al. [45] reported a low level of increase in the proline content during stress in oleander (*Nerium oleander*). Garcia-Capparos et al. [85] observed a peak in proline content at the 60 mM

NaCl level in the root tissues and in the control treatments in the leaves of the *Lavandula multifida* L. plant after the irrigation at different salt concentrations (0, 10, 30, 60, 100, and 200 mM NaCl). Mukarram et al. [72] revealed an upward trend in the proline concentration with increasing salt levels. They found that the concentration of proline increased about 2.2 times at a salt level of 240 mM NaCl compared to the control group.

Salt stress can affect the nutritional balance of a plant through a complex network of interactions, including restriction during the uptake and/or transport of nutrients from root to shoot [3]. In general, in ornamental plants grown under saline conditions, a decrease in the concentration of N, P, K, and Ca in leaf tissues was observed, while the Na and Cl concentration increased due to the antagonistic interactions associated with Na and Cl. Plant behavior regarding nitrogen uptake can differ widely under conditions of salt stress. While nitrogen uptake under salt stress may decrease, usually due to antagonism between $NH_4$ and Na or [86] Cl and $NO_3$ [87,88], the N content may also increase as N-containing amino acids such as proline increase in response to salt stress [66]. Different trends have also been observed in the P uptake of plants under salt stress conditions. Salt stress can reduce P availability due to the antagonism between Cl and $H_2PO_4$ [57]. While some researchers have detected a decrease in phosphorus content due to the competition between the mentioned Cl and $H_2PO_4$ [89], others have reported an increase in P due to the energy (ATP) required to transport the ions, which is more than necessary [90]. In our study, the nitrogen content of the salt-sensitive Zi.S increased up to 100 mM NaCl, while a sharp decrease was observed at higher concentrations. In the tolerant cultivar D.Za.F.I, N uptake increased as salt stress increased. Salt stress treatments did not affect the phosphorus content in both cultivars. Simon et al. [91] reported that salt stress led to decreased N and P content in *Chamaerops humilis* and *Washingtonia robusta* plants, and Navarro et al. [92] observed similar changes in *Dianthus caryophyllus*. Garcia Capparos et al. [93] stated that there was no consistent change in N and P concentrations in the root and leaf tissues of some ornamental plant species (*Aloe vera* L. Burm, *Kalanchoe blossfeldiana* Poelln and *Gazania splendens* Lem sp.) under salt stress. Jampeetong and Brix [94] determined that the N content of *S. natans* increased in leaf tissues but decreased in root tissues at 50, 100, and 150 mmol $L^{-1}$ NaCl concentrations.

The increase in toxic elements such as Na and Cl in the leaves of plants exposed to salt stress conditions caused visual damage such as tip and marginal blights, which negatively affected the decorative value of the ornamental plant [95]. The typical symptom of sodium (Na) accumulation was leaf blight, which occurred first on the oldest leaves and along the leaf margins. As the stress level increased, the leaf dried further towards the leaf center until all the tissue died. However, symptoms due to Cl toxicity typically begin at the leaf tip of older leaves and progress toward the stem as the stress level increases [96]. Although sodium (Na) is the main ion that causes toxicity related to high salinity, some plants are particularly sensitive to Cl. In our study, the Na and Cl content of the aerial parts of the sensitive Zi.S cultivar were found to be higher than the tolerant D.Za.F.I cultivar. Although increased sodium (Na) ions were detected in both cultivars as the salt stress increased, the percentage of the Na-increase was found to be higher in the sensitive cultivar Zi.S. The study found that the root Na content of D.Za.F.I was 1.5 times higher than its aerial part Na content, while the root and aerial part Na content of Zi.S was almost equal. The root Na content of both cultivars was found to be similar, but the aerial part Na content of Zi.S was 1.6 times higher than D.Za.F.I, indicating that the relatively sensitive cultivar transferred Na+ to the shoots at a higher rate compared to the relatively tolerant cultivar, which retained more Na+ in the root zone. As the salinity level increased, the concentration of Cl in both cultivars also increased. According to Cassaniti et al. [95], an increase in the Cl concentration in mature plants of Leptospermum scoparium led to a decrease in growth. Picchioni and Graham [97] also found that the increasing Cl concentration in seedlings of Crataegus opaca caused a decrease in growth. Controlling the salt concentration in the upper part of the plant by limiting the entry of salt ions from the roots and preventing their transport to the shoots is an important mechanism that ensures the survival and growth

of plants growing in salty conditions [3,98]. The retention of Na and/or Cl ions in root or leaf tissues is also important for salt stress tolerance [99,100]. In our study, in the relatively tolerant cultivar D.Za.F.I, this feature was manifested in the presence of higher Na ions in the roots than in the leaves. Similar results were expressed by Cassaniti et al. [95], with a higher ion concentration in the roots than in the leaves of *Viburnum lucidum*.

The physical and chemical similarities between K and Na allow Na to compete with K for binding sites on plasma membranes of root cells, leading to a reduced K uptake and decreased K availability for the plant [101,102]. The decrease in calcium intake is due to the replacement of Ca with Na in the cell membrane and the antagonistic interaction between Ca and Na ions, which impairs membrane integrity and selectivity and affects membrane properties [103]. In the present study, the aerial part K and Ca content of the tolerant D.Za.F.I was found to be higher than in the Zi.S, and the Ca content increased in both cultivars with salt stress. However, the Na/K and Na/Ca ratios in the aerial part were higher in the Zi.S than the D.Za.F.I. The concentration of $K^+$ decreased in *Celosia argentea* [104], *Limonium sinuatum* and *L. perezii* [105] under saline conditions. In addition, Carter and Grieve [106] in *Antirrhinum majus*, Navarro et al. [107] in *Arbutus unedo*, Simon et al. [91] in *Washingtonia robusta*, Navarro et al. [85] in *Dianthus caryophyllus*, Grieve et al. [108] in *Matthiola incana*, and Niu et al. [109] in *Rosa hybrida* found that the content of K and Ca decreased, while the content of Na and Cl increased. The ability of plant genotypes to maintain high levels of the K/Na ratio in their tissues is a key mechanism contributing to the expression of salt stress tolerance [110–113]. The conservation of Ca and K content in the plant under saline conditions helps maintain turgor status and cell membrane integrity [114]. Acosta-Motos et al. [51] reported an increase in Ca in different parts of *Eugenia* plants under salt stress. An increase in Ca concentrations in response to salt stress conditions has also been reported in other plant species such as *Vicia faba* L. and *Myrtus communis* L. [115,116]. Koksal et al. [117] emphasized that with the increase in the salt stress level in the *Hyacinthus orientalis* L. plant, Na intake increased significantly, the K content decreased, and the NaCl of 75 mM and above caused a sharp decrease in K/Na and Ca/Na ratios. In addition, Koksal et al. [118] stated that depending on the increase in salt stress, Ca, Mg, and Na concentrations increased while K decreased in both the roots and shoots of *Tagetes erecta*. In addition, in the same study, the roots and shoots of K/Na and Ca/Na ratios were found to be lower than the control at all salt levels. The researchers emphasized that the determination of these rates is important in terms of revealing the plant's tolerance level.

Excessive salinity also reduced the Mg absorption of plants [119]. In our study, the Mg content in the root and aerial part of D.Za.F.I cultivar did not change under salt stress conditions the root-Mg content in Zi.S cultivar decreased, while the Mg content in the aerial parts did not change. An increase in manganese content was found in the shoot tissues of the D.Za.F.I cultivar under stress conditions. At increasing salt concentrations, Rout and Shaw [120], who studied *Hydrilla erticillate* Esteves and Suzuki [119], who studied *Typha domingensis* and Jampeetong and Brix [94], who studied *S. natans*, all reported decreases in Mg content. Niu et al. [22] showed that the change in Mg content in *Zinnia marylandica* cultivars (Zahara Coral Rose, Zahara Fire, Zahara Rose Starlight, Zahara Scarlet, Zahara Yellow, and Zahara White) was minimal when compared to the changes in the amount of Na and Cl ions. Manganese is a very important trace element for plants and acts as an activator for different enzymes, which are involved in many biological events, such as oxidation, reduction, decarboxylation, and hydrolytic reactions in plant systems [121]. In our study, the effect of salt stress on the intake of microelements, the interaction effects of salt, and the cultivar were not important in roots and aerial parts tissues.

## 5. Conclusions

The study investigated the differences in salt tolerance between two Zinnia cultivars, Zinnita Scarlet (relatively sensitive) and Double Zahara Fire Improved (relatively tolerant). The results showed that the sensitive cultivar had high Na content, high ion leakage,

slow stomatal closure, reduced photosynthetic pigments, and decreased stomatal number under salt stress, while the tolerant cultivar showed quicker stomatal closure, early proline synthesis, maintained photosynthetic pigments, and low ion leakage (in 50 and 100 mM NaCl). Further studies can focus on understanding the differences from a molecular perspective and enhancing salt tolerance in Zinnia.

**Supplementary Materials:** The following supporting information can be downloaded at: https://www.mdpi.com/article/10.3390/horticulturae9020247/s1, Tables S1–S10: Cultivar and salinity effects, separately, on all plant parameters; Video S1: *Zinnia marylandica* D.Za.F.I opening stomata. Video S2: *Zinnia marylandica* ABA perfusion Video S3: *Zinnia elegans* Zi.S. ABA perfusion.

**Author Contributions:** Conceptualization, S.Y. and N.K.; methodology, S.Y. and N.K.; investigation, S.Y.; data analysis, S.Y.; writing—original draft preparation, S.Y.; writing—review and editing, S.Y. and N.K.; visualization, S.Y. and N.K; supervision, N.K. All authors have read and agreed to the published version of the manuscript.

**Funding:** This research received no external funding.

**Data Availability Statement:** The data presented in this study are available on request from the corresponding author.

**Acknowledgments:** Data presented in the study were obtained from the thesis of Sara Yasemin (corresponding author). Special thanks to the Çukurova University Scientific Research Projects Coordinating Office (Project No: FBA-2019-11481) for their support of this work.

**Conflicts of Interest:** The authors declare no conflict of interest.

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
