# Peer review of "Comparative Analysis of Morphological, Physiological, Anatomic and Biochemical Responses in Relatively Sensitive Zinnia elegans ‘Zinnita Scarlet’ and Relatively Tolerant Zinnia marylandica ‘Double Zahara Fire Improved’ under Saline Conditions"

_horticulturae, doi:10.3390/horticulturae9020247_

Round 1
Reviewer 1 Report
Review
The paper by Yasemin and Koksal., titled: “Comparative analysis of morphological, physiological, anatomic and biochemical responses in relatively sensitive Zinnia elegans ‘Zinnita Scarlet’ and relatively tolerant Zinnia marylandica ‘Double Zahara Fire Improved’ under saline conditions “ is original work, fitting to the scope of Horticulturae journal. The paper explores multi-trait acclimation response of two Zinnia species to salinity stress. I am impressed by the extent of analyses traits from morphological, anatomical, physiological and biochemical perspectives. The topic expands ecophysiological knowledge of abiotic stress resistance and acclimation for two important plant species for Horticultural production.
The paper is overall well composed and easy to read. Abstract is concise and informs the reader about most important findings of the paper. Introduction gives overview of the problematics and authors state their goals in the last section. Used methods are well described in great detail. My main concern is that authors describe “chlorophyll luminescence efficiency”, which I never heard of. The PAM fluorometer is measuring OJIP fluorescence curves, NPQ quenching and light curves. Which protocol did authors use? Authors present in Figure 7 PSII trait with μmol m-2 s-1 unit, is that basal (Fo) or maximal fluorescence (Fm)? I suggest authors to replace the trait with quantum yield (Fv/Fm) or other standard parameters used from fluorescence measurements. Visual presentation of results is clear and informative. Try to avoid broken figures or tables across multiple pages (Figure 7, Table 4) in the final revision. Authors discuss their results extensively and compare their results with other studies. The discussion enhances the overall value of the paper. Conclusion reflects the finding and is a great summary of the paper. I am looking forward to other great works of the authors. I recommend minor revision of the paper to Horticulturae. Please find my comments below:
Abstract
Line 26: PSII is not a parameter but a structure.
Line 28: I think stomatal densities is proper form and stoma is not used in English. Stomatal density is also better for the indexing of the abstract as it is widely used term in the field of botany.
Introduction
Authors could add few sentences about ABA, and its effect on abiotic stress acclimation, to explain its importance to general reader. Another suggestion is that author could mention that changes in stomatal morphology (size, density) can also improve drought resistance of plants under salinity stress.
Line 53: Please include recent papers which also reported acclimation of stomatal morphology to abiotic stress: https://doi.org/10.1016/j.plaphy.2020.11.043, https://doi.org/10.1186/s12870-021- 03304-y.
M&M
Line 148: Please expand the section and include which protocol you used (OJIP, light curves …). Were the samples dark adapted? I again suggest author to use Fv/Fm parameter if possible.
Line 230: Did you test the assumptions of ANOVA? Normal distribution of data and homoscedasticity?
Discussion
I suggest authors to include recent paper by Mukarram et al. 2022 (https://doi.org/10.3389/fpls.2022.903954) in their discussion, which also explores physiological and biochemical acclimation of lemongrass to salinity stress.

Author Response
Dear Reviewer
We would like to express our gratitude for the valuable feedback and contributions provided in the review process of our manuscript. Through your efforts, the accuracy and utility of our data, as well as the discussion and introduction, have been greatly improved. Your suggestions regarding PSII and the literature recommended for ABA and stomatal morphology have been particularly instrumental in enhancing the reliability of our results. We are appreciative of your time and expertise.
Abstract
1- REVIEWER: Line 26: PSII is not a parameter but a structure.
AUTHOR: After your revision, We checked carefully the our experimental data. We measured with the FluorPen only minimum fluorescence (Fo’). Due to the some technical problems, We could not measure PSII maximum efficiency (Fv’/Fm’) or variable fluorescence (Fv’). In this case I adjusted all PSII results according to Fo’ (minimum fluorescence) in abstract, material method, results and discussion. Thank you for this important correction.
2- REVIEWER: Line 28: I think stomatal densities is proper form and stoma is not used in English. Stomatal density is also better for the indexing of the abstract as it is widely used term in the field of botany
AUTHOR: revised
Introduction
2- REVIEWER: Authors could add few sentences about ABA, and its effect on abiotic stress acclimation, to explain its importance to general reader. Another suggestion is that author could mention that changes in stomatal morphology (size, density) can also improve drought resistance of plants under salinity stress.
Line 53: Please include recent papers which also reported acclimation of stomatal morphology to abiotic stress: https://doi.org/10.1016/j.plaphy.2020.11.043 https://doi.org/10.1186/s12870-021-03304-y .
AUTHOR: revised, added in the manuscript
M&M
3- REVIEWER: Line 148: Please expand the section and include which protocol you used (OJIP, light curves …). Were the samples dark adapted? I again suggest author to use Fv/Fm parameter if possible.
AUTHOR: revised, added in the manuscript
4- REVIEWER: Line 230: Did you test the assumptions of ANOVA? Normal distribution of data and homoscedasticity?
AUTHOR: The homogeneity of variance among treatments was tested with Levene’s test.
Discussion
5- REVIEWER: I suggest authors to include recent paper by Mukarram et al. 2022
(https://doi.org/10.3389/fpls.2022.903954) in their discussion, which also explores physiological and biochemical acclimation of lemongrass to salinity stress
AUTHOR: added and revised

Reviewer 2 Report
The authors of the work "Comparative analysis of morphological, physiological, anatomic and biochemical responses in relatively sensitive Zinnia elegans 'Zinnita Scarlet' and relatively tolerant Zinnia marylandica 'Double Zahara Fire Improved' under saline conditions" studied the effect of different concentrations of NaCl on morphological, physiological, anatomical and biochemical features of two species of zinnias. According to the results of the work, it was revealed that the plants differed in salt tolerance. So Zinnia elegans ‘Zinnita Scarlet’ sensitive cultivar had a high Na content in the aerial parts compared to Zinnia marylandica ‘Double Zahara Fire Improved’. As NaCl concentration increased, leaf thickness and height of parenchyma cells decreased in Zinnia elegans ‘Zinnita Scarlet’. A relatively tolerant cultivar (Zinnia marylandica 'Double Zahara Fire Improved') showed faster stomatal closing compared to Zinnia elegans 'Zinnita Scarlet'. In addition, the same variety began to produce proline even at low salt tolerance, retained its photosynthetic pigments, and did not allow excessive absorption of Na + by the aerial parts of the plant.
The authors have taken a lot of measurements and done a great job, but it makes little impression. It is not clear how this work differs from many others? What exactly did the authors find? In addition, the work itself is poorly written. Lots of spelling and stylistic mistakes.
English translation is bad
Throughout the text, where there is an abbreviation of the studied varieties, and somewhere their full names are given.
page 2 line 96... aimed to determine the differences in the magnitude and time of onset of the stage of salt stress between sensitive and tolerant cultivars. The task has not been completed!
Page 3 line 100 In the present study a comparative analysis was ... Why do you have a link in the purpose of the work?
Page 3 line 115. 0 concentration, is it control? Distilled water? Or nutrient medium? What was NaCI dissolved in?
Page 5 line 197.The youngest fully developed leaves (third leaf from the apex)... . So which leaves, young or fully developed?
Page 5 line 220. Why is nothing written about carotenoids in the Photosynthetic Pigment and Total Carotenoid Content method?? What was measured by this wavelength 652?
Page 5 line 230. LSD decode.
Страница 6 строка 243. How much did the percentage go down?
Why this proposal, if there is a caption under the picture? The appearance of Zinnia marylandica ‘Double Zahara Fire Improved’ (D.Za.F.I) and Zinnia elegans ‘Zinnita Scarlet’ (Zi.S) under application of saline irrigation is shown in Figure 1.
Figure 1. The appearence of Zinnia marylandica ‘Double Zahara Fire Improved’ (D.Za.F.I) and Zinnia 256 elegans ‘Zinnita Scarlet’ (Zi.S) under salt stress.
This should be removed from the additional information. Or in methods
line 260.The interaction of salinity and cultivar was statistically important in terms of stomatal parameters. It was
Why was it necessary to measure the stomata from both sides of the leaf?
Page 10 line 243-301. In the study, the opening stomatal aperture behavior of the control group plants was investigated by shining light on the closed stomata on the abaxial surface of the leaves. . What for? Why only the control group. How is this related to salinity?
Page 10 line 302-308. move everything to methods.
Page 10 line 310. The interaction of salinity and cultivar was statistically important for all leaf anatomical parameters ... . It happened before.
Page 10 line 288. In the study, the opening stomatal aperture behavior of the control group plants was investigated by shining light on the closed stomata on the abaxial surface of the leaves. In Figure 3, sample views in two cases with stomata open and closed in D. Zahara Fire Im proved and Zinnita Scarlet cultivars were presented.
Why did it have to be done? What plants did it do, saline or control?
Page 10 line 302.
The closing times of the stomata were observed after perfusing ABA (abscisic acid) from the leaf surface (intact) on the plant in the Double Zahara Fire Improved and Zinnita
Scarlet cultivars. After ABA perfusion, the stomas of D. Zahara Fire Improved closed more quickly than Zinnita Scarlet (Figure 4b). Videos of closing stomatal apertures can be found in Supplementary Data Video 2 (D. Zahara Fire Improved) and supplementary Data Video 3 (Zinnita Scarlet). After perfusing a solvent of the ABA solution, as a control, the stomata were not closed. This process was carried out to support the ABA effect.
Why is it in the text?
Page 10 line 310. The interaction of salinity and cultivar was statistically important... This sentence is repeated many times.
Page 12 line 327. Figure 5. palizat parenchyma, sp: spongy parencyma, mistakes in words
hair is not hair, this is Trichomes
Page 13 line 383Loss of turigidity was observed to increase in both cultivars, but interaction effects did not differ between cultivars Что это значит??
Ion leakage was lower in 100 mM NaCl in D.Za.F.I than Zi.S. While Zi.S. showed a dramatic increase of ion leakage under 100 mM NaCl concentration, Double Zahara Fire Improved. What did you want to say?
Photosynthesis rate (PSII) were found to affect salinity and cultivar interaction. It's not clear, how is it?
Photosynthetic pigments were also affected by asalinity and cultivar interaction. It's not clear, how is it?
Page 14 line 327.Figure 7H . Error in figure caption. Why was it necessary to make chlorophylls and total chlorophyll separately?
Page 15 line 371. Interaction effects of cultivar and salt stress on plant nutrient elements (N, P, K, Ca, Mg, Fe, Cu, Mn and Zn) in aerial parts of both cultivars were shown in Table 5. It's in the table name.
with salinity (up 100 mM) in Zi.S.. extra points
Page 16 line 383. Figure 8. Root (%) and aerial part (%) what percentage??? Why do the contents of Na and CI differ greatly in the control in the roots and in the aerial part?
Page 17 line 396. This excessive accumulation of salt ions slows plant growth as a result of the decrease in the rate of photosynthesis and the formation of growth-in ibiting metabolites. In this study, shoot length (cm), branch number (unit), branch length (cm), stem diameter (mm), root collar diameter (mm), leaf width (mm), leaf length (mm), root fresh weight (g), and root dry weight (g)
Why are there units of measurement?
Page 17 line 420. Likewise, Gomez-Bellot et al. [48] reported link is not correct
Page 17 line 422 Vibirnum tinus. Koyro mistake in the name of the species and the link does not match.
It is known that salt stress affects photosynthesis both in the short and long term. In the short term, as a result of stoma limitations, a decrease in the rate of assimilation occurs. What?
This effect can stop plant growth even after several hours of exposure to salt.
Chlorophyll deterioration caused by salt stress has been reported in many scientific studies. In general, PSII yield decreases in plants under salt stress. What is it like?
Many studies have shown that Photosystem II decreases with salt stress. The sentence was written twice. PSII abbreviation

Author Response
Dear Reviewer
The authors have attentively applied the suggested revisions to the manuscript, and have provided explanations for the changes made. The identification of typographical errors and inaccuracies by the peer reviewer is appreciated and has been beneficial in improving the overall quality of the manuscript. The authors express gratitude for the time and effort invested by the peer reviewer in evaluating the submission. The authors stand ready to address any further feedback or concerns the peer reviewer may have upon re-evaluation of the revised manuscript.
1- REVIEWER: The authors have taken a lot of measurements and done a great job, but it makes little impression. It is not clear how this work differs from many others? What exactly did the authors find?
AUTHORS: In this study, I compared the response of the relatively sensitive and tolerant Zinnia cultivars to the salt stress. To find the difference responses in same physiologic, anatomic, biochemical or morphologic progress can create new questions in the researchers mind. Why one closed its stomata quickly, and why not other? Why one could produced high amount prolin in 50 mM NaCl, why not the other? Why one cultivar defenced itself by inhibition to take Na to aerial part in 50, 100 mM, the other could not? Similar questions can be for ion leakage or other parameters. This results can be highlights to create future studies. For example after this study, one of the writers has a new project about ABA overexpression studied with CRISPR on an annual ornamental plant. We detected here difference responds for the same progress in Zinnia plants
2- REVIEWER: In addition, the work itself is poorly written. Lots of spelling and stylistic mistakes. English translation is bad
AUTHORS: English editing was done by a native speaker of the institution – ILVO-Ghent/Belgium.
3- REVIEWER: Throughout the text, where there is an abbreviation of the studied varieties, and somewhere their full names are given.
AUTHORS: abbreviation revised
4- REVIEWER: page 2 line 96... aimed to determine the differences in the magnitude and time of onset of the stage of salt stress between sensitive and tolerant cultivars. The task has not been completed!
AUTHORS: The sentence which written here made misunderstanding. It was revised.
5- REVIEWER: Page 3 line 100 In the present study a comparative analysis was ... Why do you have a link in the purpose of the work?
AUTHORS: revised
6- REVIEWER: Page 3 line 115. 0 concentration, is it control? Distilled water? Or nutrient medium? What was NaCI dissolved in?
AUTHORS: revised in the manuscript
7- REVIEWER: Page 5 line 197.The youngest fully developed leaves (third leaf from the apex)... So which leaves, young or fully developed?
AUTHORS: revised in the manuscript, fully expanded should be, thank you for corrections
8- REVIEWER: Page 5 line 220. Why is nothing written about carotenoids in the Photosynthetic Pigment and Total Carotenoid Content method?? What was measured by this wavelength 652?
AUTHORS: Arnon (39) was reported the alternatively method to measure of total chl; Total chlorophyll= [(Abs652 x 1000) / 34.5] x (mL -Acetone)/Leaf (mg)
For this reason I measured all the absorbans advised in Arnon (39). Because the results are same, I used the formula that I mentioned in the paper. So i removed 652 nm information from the passage.
9- REVIEWER: Page 5 line 230. LSD decode
AUTHORS: revised
10- REVIEWER: Why this proposal, if there is a caption under the picture?
The appearance of Zinnia marylandica ‘Double Zahara Fire Improved’ (D.Za.F.I) and Zinnia elegans ‘Zinnita Scarlet’ (Zi.S) under application of saline irrigation is shown in Figure 1.
Figure 1. The appearence of Zinnia marylandica ‘Double Zahara Fire Improved’ (D.Za.F.I) and
Zinnia 256 elegans ‘Zinnita Scarlet’ (Zi.S) under salt stress.
This should be removed from the additional information. Or in methods
AUTHORS: revised
11- REVIEWER: line 260.The interaction of salinity and cultivar was statistically important interms of stomatalparameters. It was
Why was it necessary to measure the stomata from both sides of the leaf?
AUTHORS: The interaction of salinity and cultivar was statistically important in terms of stomatal parameters --- revised
Herbaceous plants typically have stomata on both the abaxial and adaxial sides of their leaves, but there are several differences between them. Stomatal density is usually higher on the abaxial surface than on the adaxial surface. Additionally, the guard cells and stomatal pores of abaxial stomata are typically larger and wider, and abaxial stomata play a bigger role in gas exchange than adaxial stomata. Furthermore, the sensitivity of stomatal movement in response to environmental stimuli varies significantly between the two types of stomata (Wang et al 1998).
In the abaxial and adaxial surface of Zinnia has stomata. In this case, we were curious about the see the differences. So we investigated bith sides.
AUTHORS: revised
15- REVIEWER: Page 10 line 288. In the study, the opening stomatal aperture behavior of the control group plants was investigated by shining light on the closed stomata on the abaxial surface of the leaves. In Figure 3, sample views in two cases with stomata open and closed in D. Zahara Fire Improved and Zinnita Scarlet cultivars were presentedWhy did it have to be done? What plants did it do, saline or control?
AUTHORS: In the 11. and 12. explanations include also this question. revisedWhat plants did it do, saline or control? Control plants; it revised also in the manuscript figure 3.
16- REVIEWER: Page 10 line 302. The closing times of the stomata were observed after perfusing ABA (abscisic acid) from the leaf surface (intact) on the plant in the Double Zahara Fire Improved and Zinnita Scarlet cultivars. After ABA perfusion, the stomas of D. Zahara Fire Improved closed more quickly than Zinnita Scarlet (Figure 4b). Videos of closing stomatal apertures can be found in Supplementary Data Video 2 (D. Zahara Fire Improved) and supplementary Data Video 3 (Zinnita Scarlet). After perfusing a solvent of the ABA solution, as a control, the stomata were not closed. This process was carried out to support the ABA effect.
Why is it in the text?
17- REVIEWER: Page 10 line 310. The interaction of salinity and cultivar was statistically important... This sentence is repeated many times.
AUTHORS: revised- In the horticultural studies, for the experimental designs with more than one factor, we are evaluating the experiment for all factor and interaction, separately. In this study we gave the interaction results to see the salt affects on cultivars, comparatively. For this reason, when I am talking about the results, I need to tell interaction results. Because the salt and cultivar results without interaction also were given in the supplementary data.
But I revised all these types sentences as you adviced.
18- REVIEWER: Page 12 line 327. Figure 5. palizat parenchyma, sp: spongy parencyma, mistakes in words hair is not hair, this is Trichomes
AUTHORS: revised
“Glandular trichomes are specialized hairs found on the surface of about 30% of all vascular plants and are responsible for a significant portion of a plant’s secondary chemistry. Virtually all plant species possess some kind of hair-like epidermal structures. When these structures are present on the aerial parts of a plant, they are commonly referred to as trichomes, while similar outgrowths from the root are called root hairs. Trichomes—the term deriving from the Greek word “trichos”, which means hair—are, in most cases, not connected to the vascular system of the plant, but instead are extensions of the epidermis from which they originate.” Gals et al 2012. Int. J. Mol. Sci. 2012, 13, 17077-17103; doi:10.3390/ijms131217077.
As many studies mentioned trichomes is a type of glandular hair. With reviewer advise and these reports, I revised the glandular hair – trichome in the manuscript.
19- REVIEWER: Page 13 line 383Loss of turigidity was observed to increase in both cultivars, but interaction effects did not differ between cultivars Что это значит??
Ion leakage was lower in 100 mM NaCl in D.Za.F.I than Zi.S. While Zi.S. showed a dramatic increase of ion leakage under 100 mM NaCl concentration, Double Zahara Fire Improved.
What did you want to say?
AUTHORS: revised in the manuscript
20- REVIEWER: Photosynthesis rate (PSII) were found to affect salinity and cultivar interaction. It's not clear, how is it? Photosynthetic pigments were also affected by asalinity and cultivar interaction. It's not clear, how is it?
AUTHORS: revised in the manuscript
21- REVIEWER: Page 14 line 327.Figure 7H . Error in figure caption. Why was it necessary to make chlorophylls and total chlorophyll separately?
AUTHORS: revised
In chlorophyll concentration studies, generally total chlorophyll amount is given. In this study, total, a and b were determined and separately presented. Because chl a, b has different processes. If some readers curious about a and b, they can look interested figure, If some readers curious about only total chl, they can look only the total chl figure.
22- REVIEWER:Page 15 line 371. Interaction effects of cultivar and salt stress on plant nutrient elements (N, P,K, Ca, Mg, Fe, Cu, Mn and Zn) in aerial parts of both cultivars were shown inTable 5. It's in thetable name.
with salinity (up 100 mM) in Zi.S.. extra points
AUTHORS: revised
23- REVIEWER:Page 16 line 383. Figure 8. Root (%) and aerial part (%) what percentage??? Why do thecontents of Na and CI differ greatly in the control in the roots and in the aerial part?
AUTHORS: revised. Regulation of Na and Cl (transporter system, exclusion mechanisms of Na and Cl distribution) show differences. For this reason accumulation of these ions can be different. Also Griev and Walker reported that ‘regulation of Na+ distribution in Trifoliata is largely independent of the mechanism regulating C1- distribution.’ And they mentioned in their study ‘A similar conclusion has been drawn for soybeans (Lauchli and Wieneke 1978)’.
Grieve AM Walker RR (1983) Uptake and distribution of chloride, sodium and potassium ions in salt-treated citrus plants. Australian Journal of Agricultural Research 34, 133-143. https://doi.org/10.1071/AR9830133
24- REVIEWER: Page 17 line 396. This excessive accumulation of salt ions slows plant growth as a result of the decrease in the rate of photosynthesis and the formation of growth-inibiting metabolites. In this study, shoot length (cm), branch number (unit), branch length (cm), stem diameter (mm), root collar diameter (mm), leaf width (mm), leaf length (mm), root fresh weight (g), and root dry weight (g) Why are there units of measurement?
AUTHORS: revised
25- REVIEWER: Page 17 line 420. Likewise, Gomez-Bellot et al. [48] reported link is not correct
Page 17 line 422 Vibirnum tinus. Koyro mistake in the name of the species and the link does not match
AUTHORS: revised
26- REVIEWER : It is known that salt stress affects photosynthesis both in the short and long term. In the short term, as a result of stoma limitations, a decrease in the rate of assimilation occurs. What?
This effect can stop plant growth even after several hours of exposure to salt
AUTHORS: revised in the manuscript
27- REVIEWER: Chlorophyll deterioration caused by salt stress has been reported in many scientific studies. In general, PSII yield decreases in plants under salt stress. What is it like? Many studies have shown that Photosystem II decreases with salt stress. The sentence was written twice. PSII abbreviation
AUTHORS: there were some explanation mistakes. With other reviewer’ experiences, this paragraph was revised in the manuscript.

Round 2
Reviewer 2 Report
Dear authors!
Unfortunately, not all edits have been made. Much remains unclear, for example, where are the descriptions of the results of stomata, they are not. Spelling errors not corrected. The results are poorly described. The discussion needs to be made shorter but meaningful.
Below are the remarks:
Line 60 introduction ABA no decryption
Line 121 2.2. Experimental design and treatments 3cm and a height of 4.5cm. Put a space.
Line 161 Minimum Fluorescence (FO’) (FO’) remove from title
Line 205 Investigation of Stoma Behaviors against Abscisic Acid (ABA) Perfusion. Dear author, if the name is abbreviated, then the full name is given at the first mention in the text.
Line 275 The appearance of Zinnia marylandica D.Za.F.I and Zinnia elegans Zi.S. Why full names? 278 Table 1. Effects of salinity and cultivar interaction on plant growth parameters of Zinnia marylandica ‘Double Zahara Fire Improved’ (D.Za.F.I) and Zinnia elegans ‘Zinnita Scarlet’ (Zi.S). 284 Figure 1. The appearence of Zinnia elegans ‘Zinnita Scarlet’ (Zi.S) and Zinnia marylandica ‘Double 284 Zahara Fire Improved’ (D.Za.F.I) under salt stress.
And further in the text.
Line 316 3.2.2. Stoma Behavior Against Light Application and Abscisic acid (ABA) perfusion. Abbreviation.
Line 323 Where are the results. Write it in the text what data did you get by this method?! No need to refer to the video, in the results they write about the results.
Line 355 palizat replaced by palisade
Line 362 Ion leakage had lowest value in control group of D.Za.F.I, and control group of Zi.S and 50 mM NaCl of D.Za.F.I followed. Show with numbers how much it was, how much it became.
364 The ion leakage increased drastically in 100 mM NaCl of Zi.S cultivar. How much has it increased?
Line 367 While proline content increased in the 50 mM NaCl level in D.Za.F.I, it increased in 100 mM NaCl in Zi.S. A greater proline content was found in D.Za.F.I cultivar as compared to Zi.S. (Figure 7C). Show in numbers how much the increase was or give as a percentage.
Line 375 (Figure 7E, 7F, 7G, 7H). Right so (Figure 7E, F, G, H)
Line 376 Separately the effects of cultivar and salt on leaf anatomical parameters were given in Table S7 and S8. Suggestion to remove into morphology
Line 379 Error in the signature, it is necessary to write carotenoids, and you have written caroteanoid
Line 391 The results showed that there were no statistically significant… Authors, you start each paragraph in the results with this phrase. The fact that the results obtained are statistically significant is clear.
Line 393 However, it was observed that the N content in the cultivar D.Za.F.I increased with increasing salinity, while it dramatically decreased with salinity levels greater than 100 mM in the cultivar Zi.S. Show changes in numbers. if you write increased by so much ...., give the numbers.
Line 516-517 Carotenoids are one of the important biochemical substances that determine the tolerance level of the plant in case of stress. Carotenoids are important pigments that also act as antioxidants that protect membrane lipids against oxidative stress caused by stress in plants exposed to salt stress [78, 79]. These sentences have the same meaning.
Line 519 Understress conditions, violaxanthin, a carotenoid, is converted to zeaxanthin by the violaxan thin de-epoxidase enzyme [80]. Zeaxanthin has also been reported to play an important role as an antioxidant [81]. Violaxanthin, zeaxanthin did not change why discuss it? Remove this suggestion.
Line 535 Salvinia auriculata Aubl. This aquatic plant is not correct to compare terrestrial plants with aquatic ones.
Line 541 They found that the concentration of proline increased by 122%. Remove percentages and write how many times increased (1.5 or 5 times).
Line 581 In our study, the Na and Cl content of the aerial parts of the sensitive Zi.S cultivar were found to be higher than the tolerant D.Za.F.I cultivar. Although increased sodium (Na) ions were detected in both cultivars as salt stress increased, the percentage of increase was found to be higher in the sensitive cultivar Zi.S. While in the aerial part tissues, the Na+ increase percentages were 78%, 171%, 892% and 1135%, in 50, 100, 150 and 200 mM NaCl treatments, respectively, compared to the control in D.Za.F.I, it was 263%, 880%, 1238% and 1355% in the Zi.S cultivar. This is removed from the discussion and transferred to the results.
78%, 171%, 892% and 1135%, percentage of what, of what value????
Line 590 terms of Cl accumulation, 62.5%, 75%, 62.5% and 87.5% 590 increases were determined in the root tissues of D.Za.F.I cultivar at 50, 100, 150, 200 mM NaCl applications, respectively, as compared to the control. Accumulation of Cl, 87.5%, 592 87.5%, 50%, and 175% increases were found in Zi.S as compared to the control. In the aerial 593 parts, at the same NaCl concentrations of irrigation water, 136%, 200%, 209% and 264% increases were determined in D.Za.F.I, while 120%, 160%, 280% and 300% Cl increases were observed in Zi.S. Write changed from and to.
Line 593 Accumulation of Cl, 87.5%, 592 87.5%, 50%, and 175% increases were found in Zi.S as compared to the control. In the aerial parts, at the same NaCl concentrations of irrigation water, 136%, 200%, 209% and 264% increases were determined in D.Za.F.I, while 120%, 160%, 280% and 300%. Write changed from and to.

Author Response
Dear Referee,
We did corrections with your advice. Thank you for your important suggestions to make our article more valuable.
Kind regards,
Referee: Unfortunately, not all edits have been made. Much remains unclear, for example, where are the descriptions of the results of stomata, they are not. Spelling errors not corrected. The results are poorly described. The discussion needs to be made shorter but meaningful.
Authors: Dear Reviewer, stomata behaviour results revised. Spelling errors were controlled from beginning till end and revised. The results and discussion were also revised with your advices.
- Referee: Line 60 introduction ABA no decryption
Authors: revised
- Referee: Line 121 2. Experimental design and treatments 3cm and a height of 4.5cm. Put a space.
Authors: revised
- Referee: Line 161 Minimum Fluorescence (FO’) (FO’) remove from title
Authors: removed
- Referee: Line 205 Investigation of Stoma Behaviors against Abscisic Acid (ABA) Perfusion. Dear
author, if the name is abbreviated, then the full name is given at the first mention in the text.
Authors: revised
- Referee: Line 275 The appearance of Zinnia marylandica Za.F.I and Zinnia elegans Zi.S. Why full names? 278 Table 1. Effects of salinity and cultivar interaction on plant growth parameters of Zinnia marylandica ‘Double Zahara Fire Improved’ (D.Za.F.I) and Zinnia elegans ‘Zinnita Scarlet’ (Zi.S). 284 Figure 1. The appearence of Zinnia elegans ‘Zinnita Scarlet’ (Zi.S) and Zinnia marylandica ‘Double 284 Zahara Fire Improved’ (D.Za.F.I) under salt stress. And further in the text.
Authors: revised
- Referee: Line 316 3.2.2. Stoma Behavior Against Light Application and Abscisic acid (ABA) perfusion. Abbreviation.
Authors: revised
- Referee: Line 323 Where are the results. Write it in the text what data did you get by this method?! No need to refer to the video, in the results they write about the results
Authors: Revised
- Referee:
Authors: revised
- Referee: Line 355 palizat replaced by palisade
Authors: revised
- Referee: Line 362 Ion leakage had lowest value in control group of D.Za.F.I, and control group ofS and 50 mM NaCl of D.Za.F.I followed. Show with numbers how much it was, how much it became.
Authors: revised
- Referee: 364 The ion leakage increased drastically in 100 mM NaCl of Zi.S cultivar. How much has it increased?
Authors: revised
- Referee: Line 367 While proline content increased in the 50 mM NaCl level in D.Za.F.I, it increased in 100 mM NaCl in Zi.S. A greater proline content was found in D.Za.F.I cultivar ascompared to Zi.S. (Figure 7C). Show in numbers how much the increase was or give as a
Authors: revised
- Referee: Line 375 (Figure 7E, 7F, 7G, 7H). Right so (Figure 7E, F, G, H)
Authors: revised
- Line 376 Separately the effects of cultivar and salt on leaf anatomical parameters were givenin Table S7 and S8. Suggestion to remove into morphology
Authors: Revised, removed into morphology.
- Referee: Line 379 Error in the signature, it is necessary to write carotenoids, and you have writtencaroteanoid
Authors: revised
- Line 391 The results showed that there were no statistically significant… Authors, you
start each paragraph in the results with this phrase. The fact that the results obtained
are statistically significant is clear.
Authors: Dear Reviewer,
This study was carried out with completely randomized design with two factor. One factor is : cultivar (with 2 levels) other factor is salinity (with 5 levels)
When doing statistical analysis, the design
A - cultivar
B – salinity
AxB: cultivar x salinity interaction
We designed like above
In this study, because we are comparing sensitive and tolerant cultivar we talked about only the results of AxB (cultivar x salinity) interaction. Because it was already known that sensitive cultivar would have lower value and in the high salinity plant parameters would decrease. Here we wanted to talk about in which parameter and in which step this interaction would have differences. In which parameter cultivar showed differences under salinity. For this reason, for all parameters, for all results we have started with the interaction of X parameter was important or not important. But also, Incase the someone could be curious about the separately cultivar and salinity results, we gave them as supplementary.
So in the line 391; the interaction were not important statistically. Salinity affected these parameters statistically, yes, we can see in the supplementary data, also cultivar effect was important, but not interaction. For these parameters, cultivars did not have different responses under salinity.
But with your advice, we revised the sentence, did more clear.
- Line 393 However, it was observed that the N content in the cultivar D.Za.F.I increased with increasing salinity, while it dramatically decreased with salinity levels greater than 100 mM in the cultivar Zi.S. Show changes in numbers. if you write increased by so much ....,
give the numbers.
Authors: revised, added the numbers.
- Line 516-517 Carotenoids are one of the important biochemical substances that determine thetolerance level of the plant in case of stress. Carotenoids are important pigments that also actas antioxidants that protect membrane lipids against oxidative stress caused by stress inplants exposed to salt stress [78, 79]. These sentences have the same meaning.
Authors: revised
- Line 519 Understress conditions, violaxanthin, a carotenoid, is converted to zeaxanthin by
the violaxan thin de-epoxidase enzyme [80]. Zeaxanthin has also been reported to play an
important role as an antioxidant [81]. Violaxanthin, zeaxanthin did not change why
discuss it? Remove this suggestion.
Authors: revised
- Line 535 Salvinia auriculata Aubl. This aquatic plant is not correct to compare terrestrial plants with aquatic ones.
Authors: revised, removed
- Line 541 They found that the concentration of proline increased by 122%. Remove
percentages and write how many times increased (1.5 or 5 times)
Authors: revised
- Line 581 In our study, the Na and Cl content of the aerial parts of the sensitive Zi.S cultivar
were found to be higher than the tolerant D.Za.F.I cultivar. Although increased sodium (Na)
ions were detected in both cultivars as salt stress increased, the percentage of increase was
found to be higher in the sensitive cultivar Zi.S. While in the aerial part tissues, the Na+
increase percentages were 78%, 171%, 892% and 1135%, in 50, 100, 150 and 200 mM NaCl
treatments, respectively, compared to the control in D.Za.F.I, it was 263%, 880%, 1238%
and 1355% in the Zi.S cultivar. This is removed from the discussion and transferred to the
78%, 171%, 892% and 1135%, percentage of what, of what value????
Authors: transferred to the results and revised
- 590 terms of Cl accumulation, 62.5%, 75%, 62.5% and 87.5% 590 increases were
determined in the root tissues of D.Za.F.I cultivar at 50, 100, 150, 200 mM NaCl
applications, respectively, as compared to the control. Accumulation of Cl, 87.5%, 592
5%, 50%, and 175% increases were found in Zi.S as compared to the control. In the aerial
593 parts, at the same NaCl concentrations of irrigation water, 136%, 200%, 209% and 264%
increases were determined in D.Za.F.I, while 120%, 160%, 280% and 300% Cl increases
were observed in Zi.S. Write changed from and to.
Authors: transferred to the results and revised
- Line 593 Accumulation of Cl, 87.5%, 592 87.5%, 50%, and 175% increases were found in
S as compared to the control. In the aerial parts, at the same NaCl concentrations of
irrigation water, 136%, 200%, 209% and 264% increases were determined in D.Za.F.I, while
120%, 160%, 280% and 300%. Write chang
Authors: transferred to the results and revised
